# Review of Growth Defects in Thin Films Prepared by PVD Techniques

**Peter Panjan [1,\*], Aljaž Drnovšek [1], Peter Gselman [1,2], Miha Čekada [1] and Matjaž Panjan [1]**

[1]   Jožef Stefan Institute, Jamova 39, 1000 Ljubljana, Slovenia; aljaz.drnovsek@ijs.si (A.D.);
      info@interkorn.si (P.G.); miha.cekada@ijs.si (M.Č.); matjaz.panjan@ijs.si (M.P.)
[2]   Interkorn d.o.o, Gančani 94, 9231 Beltinci, Slovenia
[\*]  Correspondence: peter.panjan@ijs.si; Tel.: +386-1-477-3278

**Abstract:** The paper summarizes current knowledge of growth defects in physical vapor deposition (PVD) coatings. A detailed historical overview is followed by a description of the types and evolution of growth defects. Growth defects are microscopic imperfections in the coating microstructure. They are most commonly formed by overgrowing of the topographical imperfections (pits, asperities) on the substrate surface or the foreign particles of different origins (dust, debris, flakes). Such foreign particles are not only those that remain on the substrate surface after wet cleaning procedure, but also the ones that are generated during ion etching and deposition processes. Although the origin of seed particles from external pretreatment of substrate is similar to all PVD coatings, the influence of ion etching and deposition techniques is rather different. Therefore, special emphasis is given on the description of the processes that take place during ion etching of substrates and the deposition of coating. The effect of growth defects on the functional properties of PVD coatings is described in the last section. How defects affect the quality of optical coatings, thin layers for semiconductor devices, as well as wear, corrosion, and oxidation resistant coatings is explained. The effect of growth defects on the permeation and wettability of the coatings is also shortly described.

**Keywords:** hard coating; growth defect; nodular defect; pinhole; flake; ion etching; focused ion beam (FIB); scanning electron microscopy (SEM); droplet

---

## 1. Introduction

The surface topography is an important characteristic of physical vapor deposition (PVD) thin films because it determines their functional performance in many applications. In general, numerous topographical imperfections on the surface degrade the quality of films; in particular cases they can even cause their catastrophic failure. The surface topography of thin films on a microscopic scale is determined by three preparation steps: (i) mechanical pretreatment of the substrate, (ii) substrate ion etching, and (iii) deposition process. Mechanical pretreatment, which usually includes grinding and polishing, can cause numerous irregularities (e.g., scratches, grooves, and ridges) on the surface of the substrate material. In the non-homogenous materials, such as tool steel, additional shallow protrusions are formed at the inclusions which are harder than the ferrous matrix (e.g., carbides and oxides) while shallow craters are formed at inclusions that are softer (e.g., MnS). Some of the protruding inclusions can be torn out during polishing due to the shear stresses and leaving pits behind. In addition, polishing residue can be incorporated into the substrate surface. All of these substrate irregularities directly affect the topography of the deposited thin film. The last step of substrate surface pretreatment normally includes cleaning by ion etching. This step is performed to remove impurities left in the previous pretreatment steps and particularly, to remove the native oxide layers and to chemically activate the substrate surface for improved film adhesion (formation of nucleation sites). However, in

the case of non-homogeneous substrate material (e.g., tool steels) the ion etching can induce significant surface topography irregularities (protrusions and craters) due to different etching rates of various phases. All substrate irregularities formed during mechanical pretreatment and ion etching directly affects the topography of thin films because the growing film replicates topographical features of the substrate surface. The last part of surface imperfections comes from the deposition process itself. The physical vapor deposition (PVD) techniques, such as sputtering and evaporation, are line-of-sight processes, meaning that the material is transferred in a straight path from the source to the substrate. Due to shading (angle-of-incidence effects) of topographical irregularities on the substrate surface the thin film material is predominantly deposited in the areas which are in direct view of the sputtering or evaporation source. The result of geometrical shadowing effect is formation of growth defects (nodular defects, pinholes, pores, and other coating discontinuities). The line-of-sight deposition magnifies imperfections (e.g., foreign particles) present on the substrate surface. Even relatively small imperfections of several tens of nanometers can grow into large micrometer-sized imperfections on the surface of the thin film. All surface irregularities introduced by the PVD deposition processes cause a significant increase of surface roughness. The minor influence on the coating roughness has a columnar structure that is also a consequence of the shadowing effect.

In the PVD literature, the term "growth defects" is used (as opposed to simply "defects") in order to emphasize that the defects result from the growth process. However, the seeds that are essential for the formation of growth defects do not originate only from the deposition process but also, and in many cases predominantly, from the substrate topographical irregularities and foreign particles on the substrate. In this review, we define the growth defect as any localized imperfection in the thin film microstructure, which is in the micrometer range and forms during the film growth regardless of the seed origin. We should distinguish the growth defects from the structural defects in the crystal lattice, such as dislocations and other imperfections in the crystal structure. These types of defects are not the subject of this review.

The type of PVD technique which is used for the preparation of thin films also plays an important role in the density of generated growth defects. The plasma-based deposition process is an intensive generator of seed particles that can induce the formation of growth defects. However, PVD techniques, such as magnetron sputtering, electron beam evaporation, cathodic arc deposition, ion beam deposition, pulsed laser deposition (PLD) and others, generate substantially different types and density of seed particles and consequently different shape, size, and density of growth defects.

However, we should emphasize that the growth defects are not only limited to PVD methods but also present in thin films prepared by other deposition methods such as chemical vapor deposition (CVD) [1–3], electrodeposition [4], plasma polymerization [5], and others (the reader interested in these methods should consult references herein).

Although the growth defects were already produced by the earliest vacuum-based deposition techniques, their systematic studies could not have been undertaken until the more advanced analytical techniques with sub-micrometer spatial resolution were available. Historically, the systematic studies of the growth defects in PVD thin films can be traced to the end of the 1960s. One of the earliest studies of the growth defects in thick metallic and oxide films prepared by electron beam evaporation was reported in 1969 by Movchan and Demchishin [6]. A few years later, in 1973, Mattox and Kominiak [7] studied nodular defects in sputtered tantalum thin film. These first studies were mainly focused on the observation of growth defect geometry by scanning electron microscopy. Rigorous scientific investigation of growth defects began in the second half of the 1970s. The mechanisms of defect growth in sputtered chromium films were reported in 1979 by Patten [8]. In the same year Spalvins [9] characterized growth defects in ion-plated copper and gold films. A few years earlier, studies of Spalvins and Brainard [10] demonstrated that nodules in thick metallic films are nucleated by substrate surface imperfections arising from asperities, pits, dust particles, and flakes.

The early studies of growth defects were concentrated on metallic films, while later research focused more on the dielectric thin films, predominantly for optical applications [11]. The research of

the growth defects in optical interference coatings for mirrors was stimulated by the development of high-power lasers, which require durable and highly reliable coatings. In many instances, optical coatings were damaged by very high intensity of the laser light. It was found that the damage mainly occurred due to nodular defects present in the coatings [11].

The earliest investigations of growth defects were followed by attempts to model defect growth and their shape. Leets et al. [5] and later Tench et al. [12] proposed a simple shadowing model to describe the geometry of the nodular defect. Their model predicted the parabolic shape of the nodular defects as was commonly observed in experiments. Dubost et al. [2] improved such a model by including the surface reaction probabilities of the depositing species. In addition to simple geometric models, several two-dimensional computer simulations [11–13] have been made based on a model by Dirks and Leamy [14]. Using this model, the authors reproduced the parabolic shape as well as the open boundaries between a nodular defect and the coating matrix. In the beginning of the 1990s, Liao et al. [15] and later Müller-Pheiffer et al. [16] upgraded the two-dimensional model with an atom surface diffusion and desorption.

The shape and inner structure of a growth defect was initially investigated by scanning electron microscopy (SEM) on the metallographic cross-sections and on fracture cross-sections. In this way, only a few growth defects could be analyzed since the observation of defects depended on sheer luck. Deeper knowledge on the internal structure of individual defects came with a focused ion beam (FIB) technique used together with SEM [12–17]. This technique allowed precise cross-section of an individual growth defect to examine its internal structure and to identify the seed for its formation. In recent years, this technique has been widely employed for studying the growth defects [18–22].

We should also mention the research in which the growth defects were intentionally produced by dispersing seeds, such as microspheres, on the substrate. These microspheres were subsequently coated by a single or multilayer thin film. Such studies enabled more systematic investigations under controlled growth conditions. One of the first experiments using microspheres was performed by Brett et al. [23]. They created nodular defects by coating polystyrene latex spheres of well-defined size ranges. Poulingue et al. [24] used diamond and silica seeds of micrometer size. Wei et al. [25] investigated the behavior of artificial nodules which were created from much smaller gold and $SiO_2$ nanoparticles. Similar investigation was performed by Mirkarimi et al. [26] using gold nanospheres of defined sizes. Recently, Cheng and Wang [27] investigated defect-driven laser-induced damages in high-reflection optical coatings using silica microsphere on substrates.

In general, the growth defects are not desired because they degrade the performance of the thin films. The detrimental effects of growth defects have been explored for a wide range of thin film functional properties. As mentioned before, one of the earliest motivations for the study of growth defects were quality problems of optical thin films and thin films in semiconductor devices. Particle contamination is especially critical in semiconductor thin film device manufacturing because it reduces the production yield of such devices as well as their performance and reliability [28,29].

There have been extensive studies on the role of growth defects on the performance of coatings for the protection against wear, erosion, corrosion, gas permeation, and others. The growth defects cause serious problems for the protective functionality of the coating because they present starting points for an environmental attack. A control over the growth defects is therefore crucial for a high-quality protective coating. When studying corrosion resistance of protective coatings, for example, Korhonen [30] demonstrated that the growth defects are the main locations for the start of pitting corrosion. In the later stages of the corrosion, such points can result in the removal of a large part or the entire coating. In the past two decades several papers have been dedicated to this topic and are described in an excellent review by Fenker et al. [31] and other papers [32–36]. Resistance of protective coatings against oxidation [37] is also a phenomenon where the growth defects act as shortcuts for oxygen diffusion. The pinholes created at pits on the substrate surface and those left by the removed nodular defects are the entrance points for molecules of the environmental species towards the substrate surface. Hence, growth defects have a very important role in gas barrier coatings too [38].

In the last two decades, growth defects have also been studied in relation to the tribological performance of the coatings [39–43]. For example, Fallqvist et al. [40] showed that the as-deposited coatings, prepared by the cathodic arc, result in a significantly higher coefficient of friction as compared to the post-polished coatings. The effect of arc-evaporated droplets on the wear behavior was investigated by Tkadletz et al. [41]. They found that droplets contribute to coating degradation by providing nucleation sites for shear cracks and by the release of abrasive fragments into the sliding contact. The particularity of the approach of our research group was that we pinpointed selected defects on the coating surface and then followed them through the tribological test with scanning electron microscopy (SEM) and a focused ion beam (FIB) microscope [42,43].

The main goal of this paper is to give an overview of the decades-long research of growth defects in thin films prepared by the PVD techniques. In this review, we cover growth defects in several different deposition techniques and their influence on various thin film applications. However, we will mainly focus on the growth defects in PVD hard coatings for protection of tools and other manufacturing components, although we will also touch upon other areas of growth defect studies, particularly in optics and microelectronics. Thermionic arc ion plating deposition system BAI 730M (Oerlikon Balzers, Balzers, Liechtenstein) [22] was used for deposition of TiN and CrN single layer and TiN/CrN double layer hard coatings. Part of the samples was coated in the cathodic arc ion plating deposition system AIPocket (Kobelco, Kobe, Japan), which is equipped with superfine cathode sources. Magnetron sputter deposition system CC800/7 (CemeCon, Wurselen, Germany) [42] was used for deposition of TiN and TiAlN hard coatings. The magnetron sputtering technique was used also for deposition of nanolayered TiAlN/CrN hard coatings, as well as for deposition of TiAlN/a-CN and TiAlN/Al2O3 double layer hard coatings (CC800/9 ML, CemeCon, Wurselen, Germany). Tool steel materials (D2, H11, L2, PM steel grade ASP30) and cemented carbide (hard metal or HM) were used as substrates. The substrates were ground and polished to a mirror-like finish with a surface roughness of $S_a$ = 0.02 µm. Before deposition, the samples were cleaned in detergents and ultrasound, rinsed in deionized water and dried in hot air.

The paper is organized as follows. We first discuss surface irregularities induced by substrate pretreatment and deposition process, continue introducing the classification of the growth defects and then discuss their origin and morphology in detail. In the last chapter we explore the influence of growth defects on the functional properties of PVD coatings.

## 2. Surface Irregularities from Substrate Pretreatment

Substrate surface preparation is an integral part of any PVD film deposition process. In practice, a pretreatment of the substrate surface is always carried out before the deposition of thin films. Normally, three stages of the substrate pretreatment are included: mechanical pretreatment (grinding, blasting, polishing), wet chemical cleaning in an ultrasonic bath, and ion etching in the vacuum chamber. Mechanical pretreatment and ion etching of substrates can induce different topographical irregularities, which cause (due to the shadowing effect) the formation of numerous small- or large-scale growth defects during the deposition of coating. We will discuss here only the surface irregularities in tool steels, since these are very commonly used substrate materials in PVD production of hard coatings. Tools steels are composed of a ferrous matrix and several types of micrometer-sized carbides, which improve their mechanical properties. In addition to carbides, non-metallic inclusions (e.g., oxides, sulfides, silicates) are also present in the tool steels as inevitable impurities. Figure 1a shows a backscattered SEM top view image of the surface of D2 tool steel after middle frequency (MF) and intensive (booster) ion etching and after deposition of nanolayer nl-TiAlN/CrN hard coating (Figure 1b). Different carbide and non-metallic inclusions as well as pits of various sizes and shapes were identified by energy dispersive X-ray analysis (EDX). Their positions as well as the positions of two relatively large but pointed marks (e.g., Vickers indentations) made at opposing sites of the sample were saved in the microscope's coordinate system. The local (sample) coordinate system was uniquely fixed at these two points. By transforming the local (sample-based) coordinate system to the current instrumental

coordinate system we are able to find the precise positions of selected inclusions on the substrate surface after deposition of coating (Figure 1b). EDX inspections were performed over five areas with equal dimensions of 158 μm × 118 μm on two different types of substrates (D2, powder metallurgical (PM) ASP30). Table 1 shows the average number of such inclusions per mm$^2$ in different types of steel. In the following sections, we will examine the influence of all substrate pre-treatment steps on the formation of large- and small-scale surface irregularities.

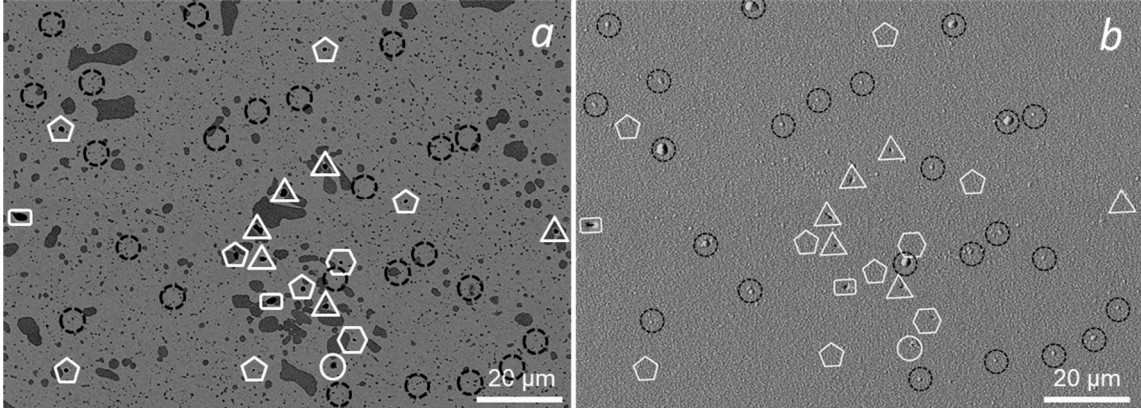

**Figure 1.** Scanning electron microscopy (SEM) images of D2 tool steel substrate after intensive (booster) and middle frequency (MF) ion etching (**a**) and the same surface area after sputter deposition of nanolayer nl-TiAlN/CrN hard coating (**b**). The sulfide, oxide, and other inclusions are marked with triangles, pentagons, and hexagons, respectively. Pits are designated with rectangles while sites where all other growth defects formed during the deposition process are labeled with black circles.

**Table 1.** Surface densities of the non-metallic inclusions in D2 and ASP30 PM tool steel substrates.

| Steel Type (AISI) | Oxide Inclusions Density (mm$^{-2}$) | Sulfide Inclusions Density (mm$^{-2}$) | All Inclusions Density (mm$^{-2}$) |
|---|---|---|---|
| D2 | 800 ± 300 | 460 ± 80 | 1400 ± 200 |
| ASP30 PM | 200 ± 120 | 5100 ± 500 | 5400 ± 300 |

*2.1. Mechanical Pretreatment*

Mechanical pretreatment of tool steels usually includes grinding [44] and polishing [45]. First, the surface is ground with progressively finer grinding papers and then it is polished with diamond paste from 15 μm down to 1 μm. There is no general recipe for polishing all types of tool materials. The grinding and polishing steps have to be slightly adjusted for each specific type of tool steel. Important parameters which determine the polishability of tool steels are the homogeneity of the microstructure, the level of purity, and the size and distribution of carbides and nonmetallic inclusions in the ferrous matrix. The biggest problems are caused by inhomogeneities. Both the purity and the homogeneity are significantly influenced by the manufacturing process of tool steel.

Although mechanical pre-treatment substantially smooths the substrate surface, it also creates numerous irregularities of the micrometer size at the same time. Mechanical pretreatment, even if performed carefully, creates various irregularities in the shape of scratches, pits, ridges, and other shapes. In addition to these relatively large topographical irregularities, smaller ones are also formed. In tool steels, slightly shallow protrusions with step-like edges are formed at the inclusions harder than the ferrous matrix (e.g., carbides and oxides) (Figure 2). At the inclusions softer than the steel matrix (e.g., MnS), inverse geometrical features are formed, known as shallow craters. The height of the protrusions and the depth of the craters depend on the hardness of the inclusions (Figure 3).

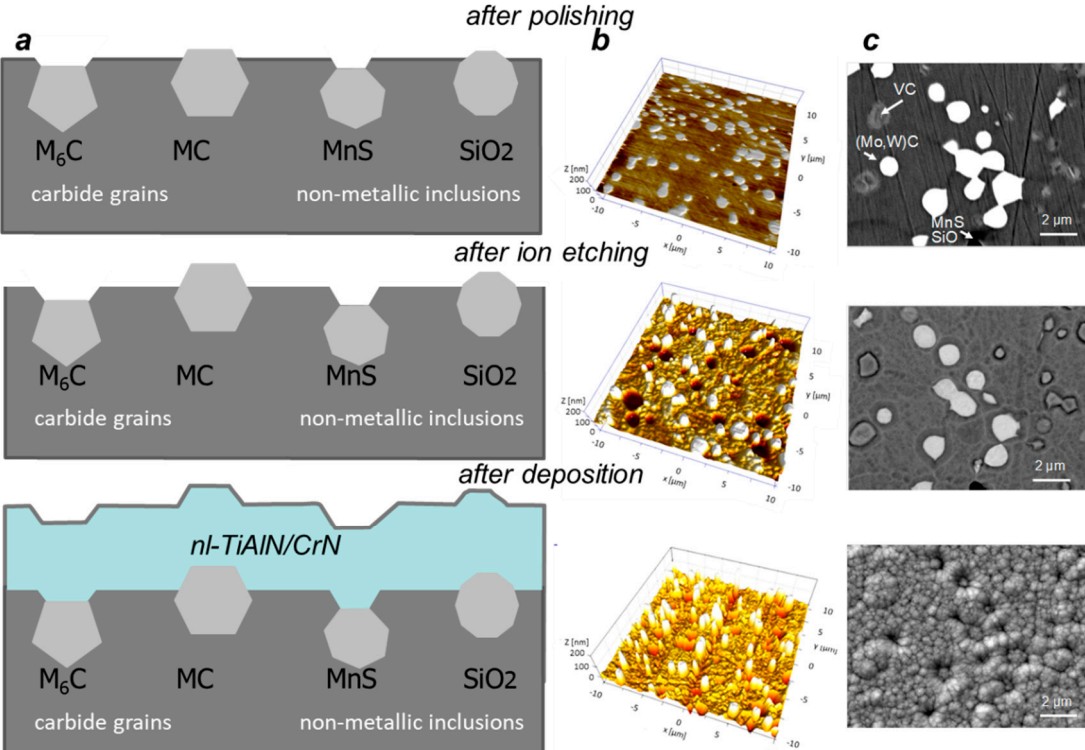

**Figure 2.** The schemes on the left side (**a**) illustrate the topographical changes of different types of inclusions during various steps of coating preparation (polishing, ion etching, deposition). Atomic force microscopy (AFM) (**b**) and SEM (**c**) images show the topography changes of ASP30 PM steel substrate after polishing, ion etching, and deposition of nl-TiAlN/CrN hard coatings. The SEM images (**c**) were taken at the same site on the substrate surface.

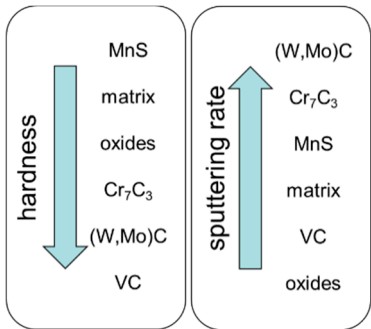

**Figure 3.** The total geometrical extension from the matrix level (either positive or negative) depends both on the differences in the polishing removal rate (hardness) and ion etching (sputtering) rate.

When polishing tool steels, a very common problem is over-polishing, when the polished surface gets rougher with the polishing time. Over-polishing is associated with two surface phenomena: "orange peel" and "pitting". The "orange peel" is a term used for a surface with randomly distributed smooth valleys and hills, which resembles the surface of an orange (hence the name). The formation of an orange-like surface is related to the clustered distribution of carbides in tool steel which can occur in the last polishing step if polishing is performed at too high of a pressure and prolonged time. In general, high hardness materials are less sensitive to this problem.

The most problematic issue of mechanical pretreatment is the formation of pits in the substrates (Figure 4). High pressures that are present during grinding and polishing can cause the formation of small pits (or cavities) at the positions of hard inclusions; the effect is therefore referred to as the pitting

effect. The high shearing stresses also present during the polishing can tear out some of the protruding inclusions and pits with dimensions of the inclusions are left behind. The best way to avoid the orange peel and pitting effects is to keep polishing pressures constant and not too high. It is also important to use short polishing steps and apply cleaning of the substrate surface after each step.

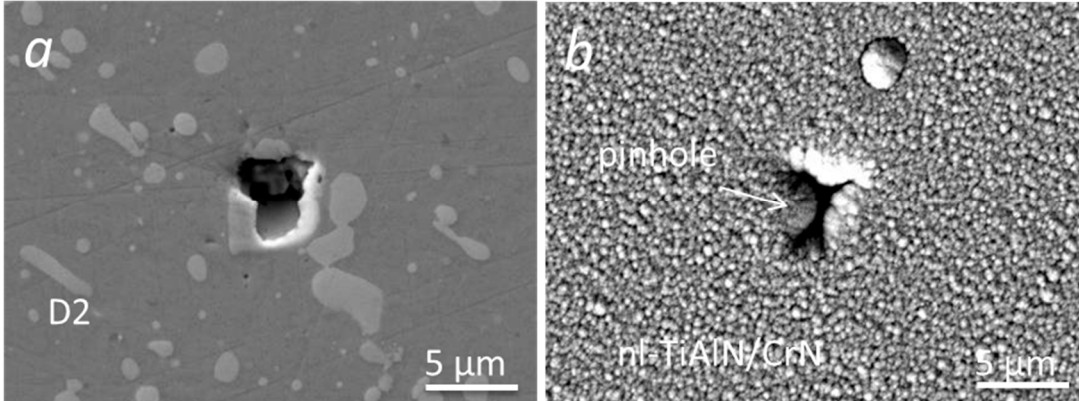

**Figure 4.** Pit on D2 tool steel substrate surface at the site where a carbide grain was torn out from the surface during the polishing (**a**) and the same substrate surface area after deposition of nl-TiAlN/CrN nanolayer (**b**) [46]. Close to the pinhole a nodular defect was also formed.

A part of mechanical pretreatment often includes dry or wet microblasting. In dry microblasting, abrasive media (e.g., corundum) is blown with compressed air onto the substrate to clean the substrate surface. Wet blasting is similar to dry blasting where abrasive media is mixed with water to form slurry. Both blasting techniques are used not only to clean the substrate surface, but also to alter the microtopography, hardness, and residual stresses of the substrate surface. In the pretreatment of cutting tools, wet blasting is also used for rounding the cutting edge. After such treatment the cutting edge has a more stable form, while the performance of cutting tools is significantly increased. It is clear that the impingement of micro-sized hard particles at high velocities can also cause numerous surface irregularities on the substrate surface.

## 2.2. Wet Chemical Cleaning

The mechanical pretreatment is always followed by the chemical cleaning of the substrates. The properties of thin films deposited by different PVD techniques depend on the cleanliness of the substrate surface on which the film is deposited [47]. Namely, any kind of contamination on the substrate surface can result in reduced adhesion of the film to the substrate, more rapid degradation of the film, greater contact resistance for electrically conducting films, and poor optical qualities for optical films. Thus, the precondition for the achievement of good adhesion of PVD coating is cleanliness of the entire substrate surface. The cleaning procedure takes place both outside (e.g., chemical cleaning) and inside (e.g., ion etching) of the vacuum chamber, just prior to the thin film deposition process.

A typical ultrasonic aqueous batch cleaning process consists of three steps: (a) ultrasonic washing in alkaline cleaning agents (pH~11); (b) ultrasonic rinsing in pure water; and (c) drying in pure hot air [47]. If we do not provide regular maintenance of water filters, air filters, and cleaning agents, then the cleaning device can also be the source of the particles, which cause the formation of growth defects. In rare cases, when the substrates are not immediately placed in the vacuum chamber for the deposition, a pitting corrosion can occur at the surface. After coating deposition growth defects are formed on the corroded area of the substrate (Figure 5).

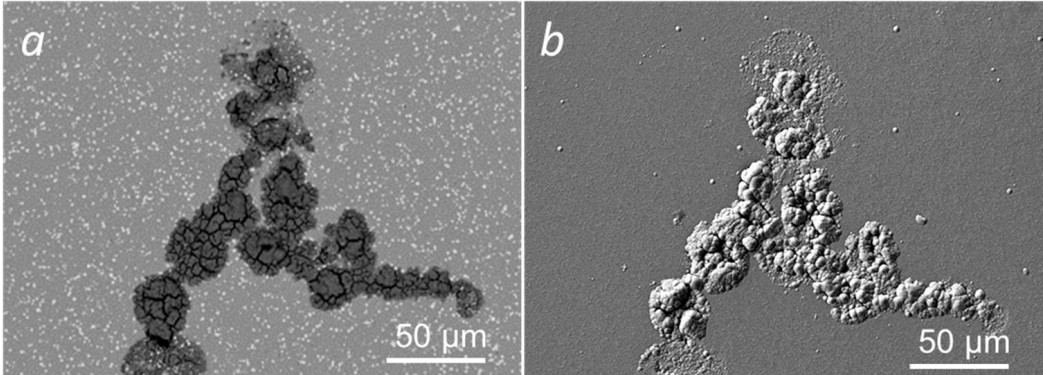

**Figure 5.** SEM images of a well-defined network of cracks on the corroded area of ASP30 PM tool steel substrate after wet cleaning and MF ion etching (**a**) and the same area after deposition of nl-TiAlN/CrN nanolayer hard coating (**b**).

## 2.3. Ion Etching

### 2.3.1. Basics of Ion Etching

The ion etching changes the microchemistry, surface topography, and the microstructure of the near-surface layer [47–49]. All these changes affect both the adhesion of the coating as well as its growth. However, an improved adhesion is not only a consequence of removal of surface oxides and other contaminants (which decrease the interface strength), but it is also a result of an increased density of nucleation sites and chemical activation of the surface layer. Substrate surfaces which are exposed to an ion etching erode and change topography. In general, surface topography depends on the duration of etching, density of the plasma, energy of ions, and type of ions. Different morphological features like cones, pits, hillocks, and pyramids are formed and their formation is closely related to the initial surface irregularities, impurities, and variations in the sputtering yield as a function of the angle of the ion beam incidence to the surface.

In PVD systems, there are two general concepts of ion etching procedure, which are schematically shown in Figure 6. The first and the simplest ion etching configuration is by generating plasma on the substrates themselves (Figure 6a). Such an ion etching procedure works only if high-frequency oscillatory voltage is applied to the substrates. In practice, middle frequency (MF) with several hundred kHz or radio frequency (RF) with 13.56 MHz is used. An advantage of this type of ion etching is that it forms globular plasma around the substrates and results in a more-or-less uniform etching of substrates. The disadvantage of such ion etching is that it does not allow independent control of ion densities and energies. This is determined by the voltage frequency and amplitude, which also affect the self-bias voltage.

In the second concept of ion substrate etching, an auxiliary plasma source is used. Auxiliary plasma can be generated in different ways therefore this type of ion etching is specific to a particular PVD technique. For example, plasma can be generated with the help of a hollow cathode (Figure 6b), a thermionic arc source (Figure 6c), a heated tungsten wire (Figure 6d), or some other type of plasma source. The auxiliary plasma is normally spatially confined and not all substrates are immersed in the plasma at the same time. For this reason, the substrates need to be rotated in the vacuum chamber and around their axes to achieve a more-or-less uniform ion etching. However, the main advantage of auxiliary plasma is that ion density can be controlled by the plasma source, while the ion energy is controlled by bias potential on the substrates. Substrates can be biased either by continuous pulsed or oscillatory potential, which allows an even wider control over the ion etching procedure. Although continuous substrate bias provides the most intense substrate etching, a pulsed or oscillatory substrate potential is still used in many cases because it enables the removal of native oxide and other non-conductive contaminates. Auxiliary plasmas can also be a source of metal ions. Such ion etching

is available in the PVD processes that have sources of highly ionized metal plasma, such as in cathodic arc deposition [34] or high-power impulse magnetron sputtering (HIPIMS) [48]. In these cases, the source of metal ions is the cathode targets themselves, which are used during the coating deposition process. As opposed to coating deposition, the etching with metal ions is performed by applying a high negative bias potential to the substrates (typically several hundred volts); in most cases direct current (DC) bias is applied. The metal ions have a higher atomic mass than the argon ions and are more efficient in the etching of the substrate materials. The metal ions do not only etch the substrates, they are also implanted in the near-surface layer of the substrate. Such a metal ion implanted interlayer is normally beneficial since it improves adhesion of the coating. A disadvantage of etching with metal ions generated by arc discharge is the contamination of substrate surface with droplets, which reduce the adhesion of the coating and cause the formation of growth defects. This disadvantage is eliminated when HIPIMS discharge is used to generate the metal ions for the ion etching [48]. In contrast, when argon ions are used for etching, argon is also implanted in the near-surface layer, but this is normally not desired since argon is implanted in the interstitial positions of the crystal lattice, which normally increase stresses in the surface layer, while the crystallinity of the interface is completely lost.

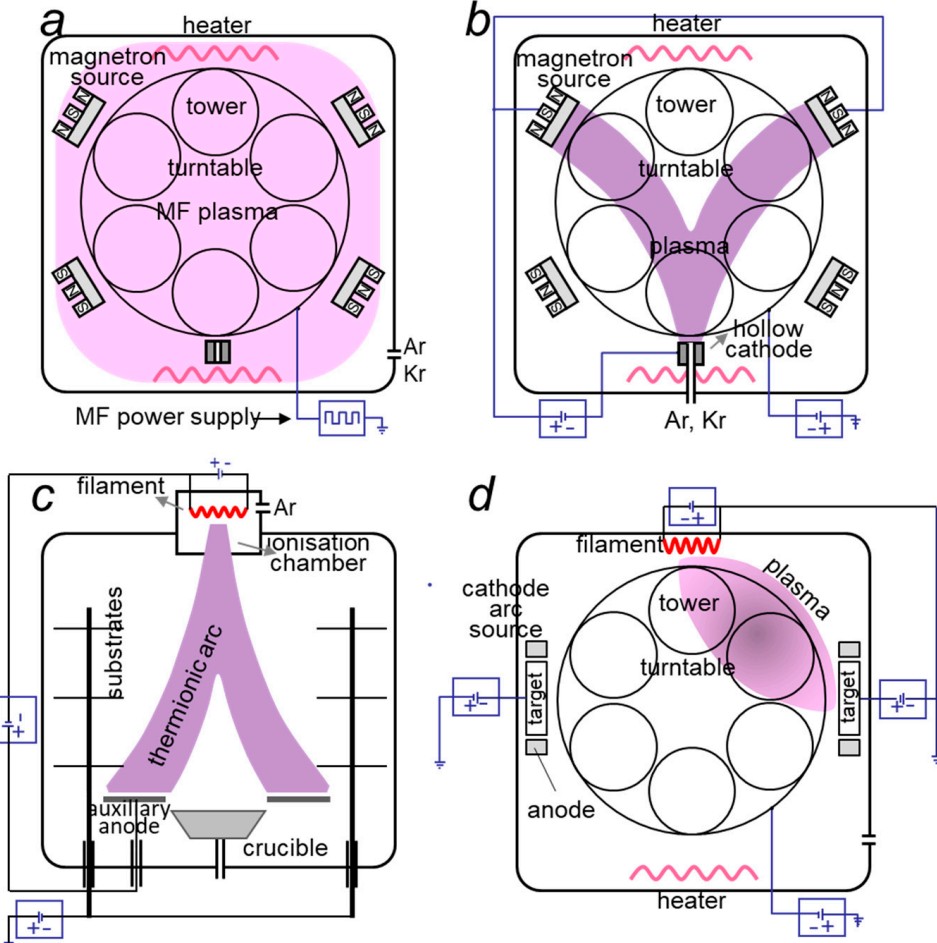

**Figure 6.** Examples of ion etching modes in the three different physical vapor deposition (PVD) systems we used for deposition of PVD hard coatings: (**a**) MF etching and (**b**) etching with hollow cathode plasma source (DC bias) in magnetron sputtering system CC800/9 ML; (**c**) ion etching (DC bias) in thermionic arc evaporation system BAI 730; (**d**) ion etching (DC bias) cathodic arc deposition AIPocket.

For ion etching of the substrate surface before deposition, inert ions from the broad ion beam sources (e.g., Kaufman ion source) can also be used. The advantages of such kinds of ion sources compared to competitive processes are that they generate an ion beam with a well-controlled direction, density, and energy.

### 2.3.2. Substrate Irregularities Induced by Ion Etching

Here we will examine substrate irregularities induced by ion etching in typical industrial PVD deposition systems we used for deposition of hard protective coatings: magnetron sputtering system CC800/9 ML, cathodic arc system AIPocket, and thermionic arc system BAI730 (see schemes in Figure 6).

If the substrate material is composed of different phases, which have different ion etching rates, then these can cause considerable geometrical irregularities on the substrate surface. An example of where such substrate irregularities form is the ASP30 PM tool steel, which is composed of several types of inclusions in the ferrous matrix. During the ion etching of tool steel material shallow craters and shallow protrusions are formed at the sites of the metal carbides and other non-metallic inclusions (Figure 2). The reason for their formation is the difference in the ion etching rate of the inclusions and the ferrous matrix. In the case of ASP30 tool steel, the sputtering rate of the $M_6C$ carbides and MnS inclusions is higher than that of the steel matrix, while that of the MC carbides and oxide inclusions is lower. The intensity of ion etching depends on the geometry of the substrate, ion current density, ion energy, rotation mode of substrate, and plasma uniformity.

The total geometrical extension from the matrix level (either positive or negative) thus depends both on the differences in the polishing removal rate (hardness) and ion etching (sputtering) rate (Figure 3). Consequently, shallow craters (on site of $M_6C$, MnS) and protrusions (on site of MC and oxides) are formed. Typical height values of protrusions and depth of craters are up to a hundred nanometers.

If a foreign particle is present on the substrate before the ion etching step, then it prevents the etching of the substrate area underneath the seed. Figure 7a shows an example of an etched substrate surface with a large irregularly-shaped foreign particle. A part the substrate underneath the seed was not ion etched. As a consequence, a step-like feature formed on the substrate surface. During deposition a nodular defect is formed at the site of such a particle. The existence of the step beneath the particle proves that it was present on the substrate surface before etching (Figure 7b). If there was no step, then it reached the surface at the end of the ion etching process or immediately after it.

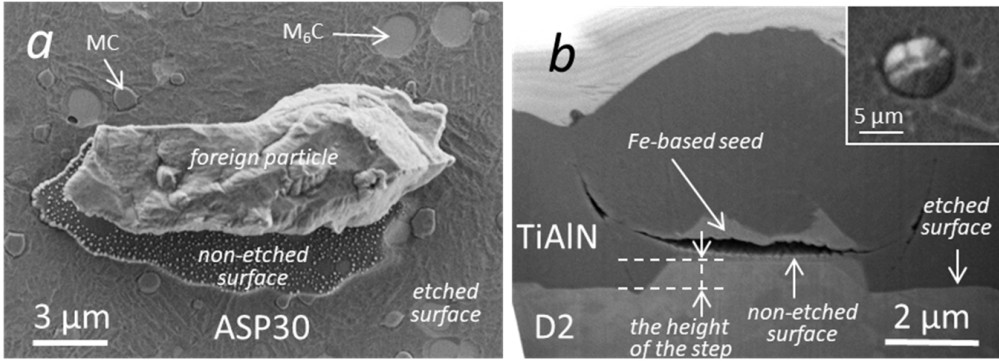

**Figure 7.** A foreign particle which remains on the substrate surface after the wet cleaning procedure, prevents etching of the substrate area underneath it (**a**); a step on the substrate surface beneath the iron-based seed proves that the seed particle was on the substrate surface even before etching (**b**).

If ion etching is not done properly then it can cause degradation of the substrate surface. Problems may arise due to the backscattering of the material sputtered from a substrate surface back to the surface, particularly at high pressure. Therefore, care must be taken to flush away the sputtered contaminant species. The same effect as backscattering is caused by redeposition which is defined as the return of the material sputtered from a substrate surface back to that surface. Another undesirable effect of ion etching in industrial deposition systems is the cross contamination of the substrate surface as well as target surface with batching material. During ion etching a part of the batching material is transported first from the substrate (tool) surface to the target surface. Later (in the early stage of deposition) the same material is returned back to the substrate surface forming a thin contamination film. To avoid contamination of the targets with batching material during ion etching, the target should be covered with moveable shutters.

## 3. Growth Defects Formed during Deposition

During deposition all morphological features of the substrate surface that are formed during its mechanical pretreatment and ion etching are transferred onto the coating surface and they are even magnified. Topographical irregularities and small foreign particles remaining on the surface of the substrate after cleaning and those which were generated during ion etching, cause the formation of growth defects in the coating due to the shadowing effect. However, a large part of the seed particles responsible for growth defect formation are generated during the coating process itself. Figure 8 shows 3D-profilometry images of a TiN coating deposited on D2 tool steel substrate by three different deposition techniques: (a) evaporation by thermionic arc (BAI730); (b) magnetron sputtering (CC800/9 ML) and (c) evaporation using cathodic arc (AIPocket). The difference in the growth defects density is evident.

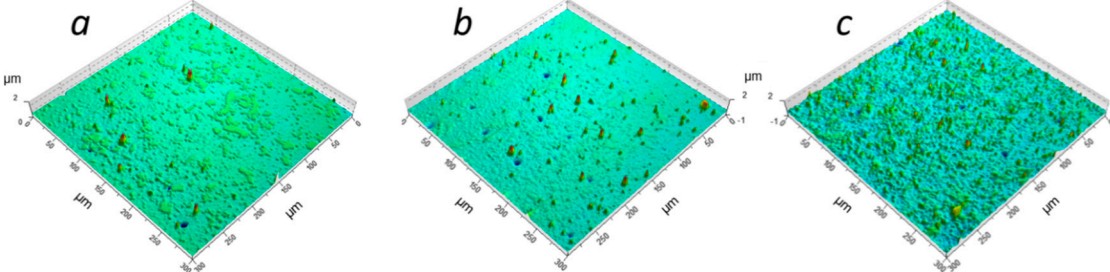

**Figure 8.** Three-dimensional (3D)-profilometry image of TiN coatings deposited on D2 tool steel substrate evaporation using thermionic arc (**a**), magnetron sputtering (**b**), and cathodic arc (**c**). The sharp peaks are the nodular defects while the blue dots are craters (pay attention to the strong exaggeration in z-scale).

In this section the growth defects according to their origin and shape are classified, while in the next one all potential sources of seed particles are described.

The literature is not consistent in classifying and naming the growth defects. The diverse classification comes from a wide variety of defect morphologies and their origins, and from different application fields studying growth defects. In this work, we propose to classify growth defects in the most general way. To keep things simple, we divide growth defects in two general groups: (i) the term "protrusions" is used for those defects that are above the mean surface of the film (Figure 9) and (ii) the term "holes" is used for those defects that are below the surface. We also attempt to provide a unified nomenclature for the defects with respect to their origin and morphology.

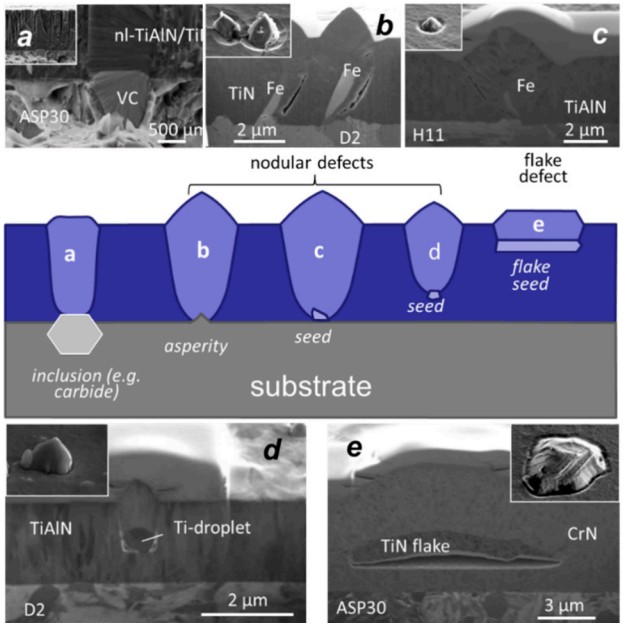

**Figure 9.** Schematic overview of different seeds causing the formation of protrusion defects: (**a**) carbide inclusion, (**b**) geometric irregularity, (**c**) foreign particle on the substrate, (**d**) foreign particle during growth, (**e**) flake. Focused ion beam (FIB) and fracture cross-section SEM images of typical protrusion defects in PVD hard coatings prepared by evaporation using thermionic arc (**b**,**c**), magnetron sputtering (**a**,**c**), and evaporation using cathodic arc technique (**d**) are added.

## 3.1. Protrusion Defects

### 3.1.1. Nodular Defects

Nodular defects are the most common type of growth defects [2,10,12,13] and they are present in all PVD coatings. The formation of nodular defects is caused by a seed. Seeds are usually very small particles (dust, foreign particles, particles ejected from the coating material source) or substrate protrusions. It appears that virtually any irregularities, even a minute one, may act as a seed. The nodules do not always start to grow at the substrate surface; they can also grow from a seed, which arrived on the substrate during the deposition process. The nodule starts to grow in the shape of an inverted cone that propagates through the film and forms a domed protrusion at the outer surface of the film. The nodule itself is much larger than the seed that causes it. It is not, in itself, a contaminant. It is composed of the same material as the coating but growing in a different way. Due to the shadowing of vapor flux by the seed particle and limited thermal mobility of atoms on the surface, there are discontinuous boundaries between the nodule and the surrounding coating matrix. The outer surface of the nodule is a quite sharp boundary between it and the remainder of the coating. This sharp boundary is a region of weakness and there is frequently an opening around the nodule, either partially or completely, and the nodule may sometimes be detached from the coating completely, leaving a hole behind.

As already mentioned, nodular defects originate from small seed particles and on the surface of a thin film appear in a shape of cones or domes (Figure 10a). In the literature, they are also called nodules, hillocks, peaks, or inverted cones. The part of the nodular defect that is below the film surface has a shape of an inverted cone with a parabolic cross-section, whereas the part of the defect above the film has a shape of a cone with a rounded top (similar to a dome). The base of the nodule dome (or hemi-sphere) is circular or oval in the planar projection (Figure 10b). Leets et al. [5] and later Tench et al. [12] developed a simple model, which described the geometry of the classical parabolic nodular defects observed experimentally. Their model was based on omnidirectional (isotropic) coating flux and on the assumptions that the nucleating particle is spherical and much smaller than the total

coating thickness. They also assumed that the growing coating has no internal structure and that the mobility of ad-atoms may be neglected. In this case, the layers of the coating material above the seed are concentric spherical caps, while the coating is perfectly conformal (the coating thickness is assumed to be identical everywhere on the seed). In this case the topology of defect growth becomes a simple geometrical problem. The relationship between the diameter of nodular defect ($D = 2R$), the seed particle diameter ($d$), and the coating thickness ($t$) for a hemispherical seed particle on a flat substrate surface is: $D = \sqrt{8dt}$. This simple geometric model explains the shape of the nodular defects but fails to explain their size (diameter and height). It is valid when the seed particles are small, while larger particles produce more complicated structures due to shadowing effects. Dubost et al. [2] improved such a model by including proposed surface reaction probabilities of depositing species.

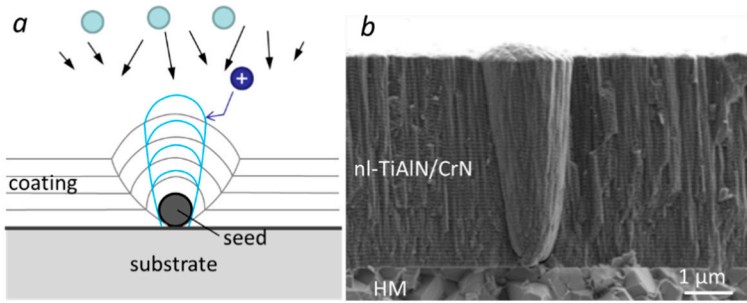

**Figure 10.** Scheme of a conical and parabolic nodule cross-section; the shape mainly depends on the flux distribution of the incoming atoms (**a**); fracture cross-section SEM image of a typical nodular defect with rather straight vertical walls in the nl-TiAlN/CrN hard coating prepared by magnetron sputtering (**b**).

In addition to geometric models for description of the nodular defect formation, several two-dimensional computer simulations [13] have been proposed. These simulations are based on hard disc model of Dirks and Leamy [14] in which discs (atoms) fall randomly onto the perfectly flat surface at a fixed oblique angle and then stick immediately where they land and roll into the nearest saddle position formed by the previously deposited discs. It should be emphasized that this computer simulation model is a purely kinematic one, because it does not take into account surface or particle energies, interatomic forces, crystallographic orientations, etc. These computer simulations show that nodular defects have a cylindrical symmetry with parabolic side wall structure and that the actual aspect ratio of the defects could be varied by changing deposition conditions in the model (oblique incidence, random variations of particle flux, rotating substrate). Using such a model they also reproduced the columnar microstructure and columnar tilt in the coatings, as well as opened boundaries between the nodular defect and coating matrix. However, the model does not consider complex adsorption processes that influence film-structure evolution. Liao et al. [15] and later Muller-Pheiffer et al. [16] upgraded the two-dimensional hard disk model in such a way that they included surface diffusion and desorption of arrived atoms. A more sophisticated ballistic model of coating growth, which takes into account scattering and surface diffusion of depositing species was developed by Lang and Xiuqin [50]. Their computer simulation shows that higher deposition rate, lower surface diffusivity of deposit, and higher degree of scattering of depositing particles favor the formation of nodular defects.

The deposition process can have a profound effect on the shape of the nodule. The nodule cross-section geometry mainly depends on the flux distribution of the incoming atoms (Figure 10). It is parabolic if the flux is random (isotropic) and conical if the flux is directional (narrow angular distribution of the deposition flux). A parabolic profile of the nodule is a characteristic for electron-beam deposition where a wide range of deposition angles is present. In sputtering, on the other hand, where narrower deposition angles are present, nodular defects with more straight vertical walls are formed.

The non-uniformity of incident angles of the incoming flux of the coating material and the shadowing effect cause the formation of nodules with asymmetrical boundaries (Figure 11a).

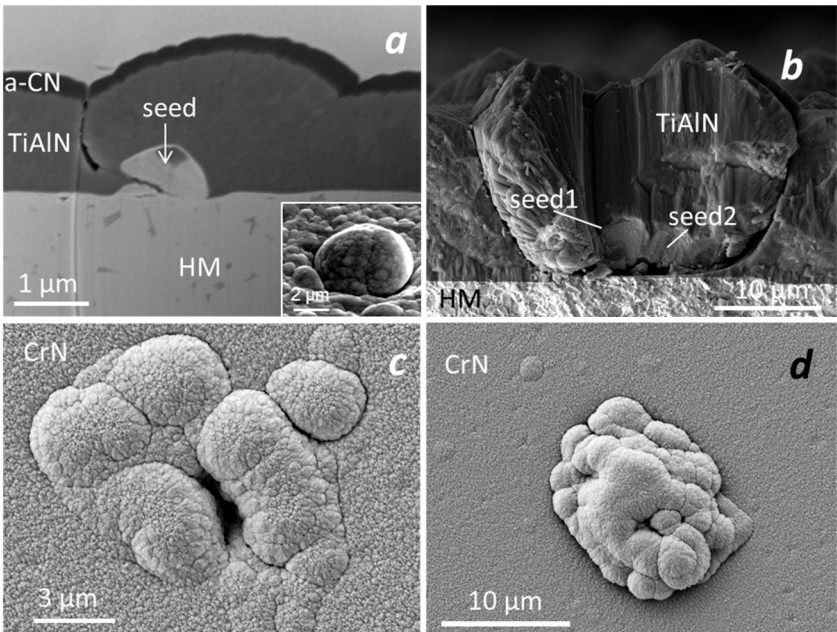

**Figure 11.** The material deposited on one seed forms a single nodular defect (**a**), while the material deposited on two or more close-spaced seeds forms conjoined nodular defects (**b**). Nodular defects can appear together in complex aggregates (**c**) or with a typical "cauliflower-like structure" (**d**). Nanolayer nl-TiAlN/CrN (**a**) and TiAlN (**b**) hard coatings were prepared by sputtering, while evaporation using thermionic arc was used for deposition of CrN hard coating (**c**,**d**).

Nodular defects forming on the substrates which rotate in their own plane have distinctly different shapes and sizes than those which form on stationary substrates [13]. Nodules on stationary substrates form an inverted cone with straight sides while those produced on rotating substrates form a rounded bowl-like bottom. In addition, for rotating substrates the nodule size and shape strongly depend on the angle of the vapor flux with respect to the substrate plane.

Nodular defects differ by their shape, seed depth, composition, and seed shape. The shape of the nodule and the structural characteristic of the nodule-coating interface are basically determined by the seed size and shape. If the seed is small in comparison to the coating thickness, then the shape of the nodular defect does not depend on the shape of the seed. If the seed has a smooth morphology then the nodule looks like a cone, while a seed with complex morphology results in the growth of a nodular defect with irregular surface features. Irregular shapes of nodular defects are not uncommon, particularly if they are very large. The nodular defects can appear individually (Figure 11a), form in clusters which can overlap each other (Figure 11b,c), or appear together in complex aggregates with a typical cauliflower-like structure (Figure 11d). When an irregularly shaped seed is coated, the particle flux cannot reach the area underneath the seed due to the shadowing of the particle flux and thus causes formation of voids below the particles. The coating on such kinds of seeds results in highly non-uniform coverage with a practically uncoated area underneath the seed. This can be seen in Figures 7b, 9a and 11b,c.

As the seed gets overgrown by the coating, the contour of the coating follows the shape of the seed. The nodular defect is a conformal replication of the seed particle shape. If the seed particle is overcoated by a multilayer coating, then the contours of individual layers show that the coating growth within the nodule continues in a similar way to the coating matrix (Figure 12a).

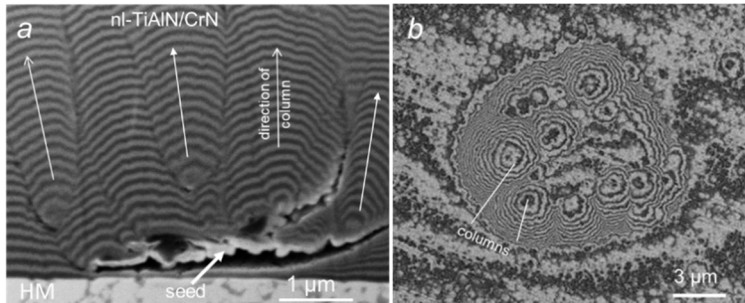

**Figure 12.** FIB image of a nodular defect in the nl-TiAlN/CrN nanolayer coating; several columns are formed on the surface of the seed with complex geometry (see arrows) (**a**) and SEM top view image of the ground section of the nodular defect caused by a seed with complex geometry (**b**).

Petrov et al. [51] argued that the growth of coating on seed particles is far more coarse and columnar compared to the regions around it because the top of the nodular defect grows in a regime of intensive ion bombardment while for seed regions in the shadow of the particle flux the ion bombardment is less intensive [52]. This has a large effect on the microstructure of the nodular defects which is composed of dense columnar grains, while the microstructure of the lower region of nodular defects is rough and underdense. The nodular defects typically grow in a "feather like" pattern growing from a central core.

Due to high internal or thermal stresses the internal cohesion of the nodular defect is often inferior to the cohesion at the boundary. In this case, only a part of the nodular defect is broken off, like the cases shown in Figure 13a,b. This may happen during deposition (high internal stresses) or during the cooling stage (high thermal stresses). The SEM image exposed the internal structure of nodular defects, which includes the microstructure of the coating as well as the size and shape of the seed particle.

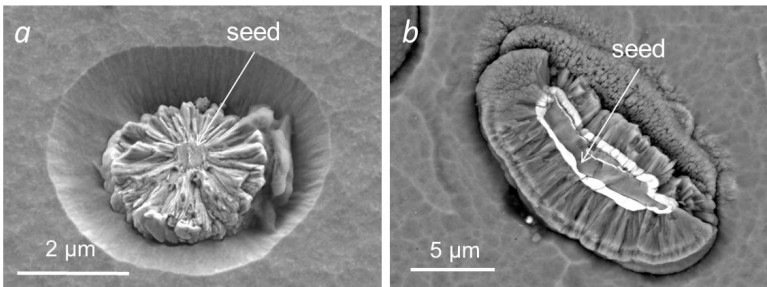

**Figure 13.** SEM micrographs of two broken nodular defects in TiAlN hard coating sputter-deposited on D2 tool steel substrate.

### 3.1.2. Flake Defects

Another type of growth defect that protrudes above the surface of a thin film is flake defects. They differ from nodular defects in the origin of the seed particles, which results in a much larger size of defects and a very different shape than nodular defects. The seed particles (see e.g., Figure 14) are large flake particles that originate from the delamination of coating that was deposited in previous batches on the fixture holders for substrate and shields within the vacuum chamber. These seed particles are delaminated due to the high thermal and internal stresses that are present in the coating prior and during the deposition. The seeds of the flake defects are typically very large, normally several tens of micrometers in the diameter and have irregular shapes. Due to very large diameter-to-thickness ratio of the flake seeds, the defects that form above the seed have flat top and step-like edges. Hence, the flake defect is a step-like projection of the overgrown seed. Flaking of the coating is also triggered by arcing on substrate fixtures (i.e., substrate turntable) during ion etching and deposition process. Overall, the surface density of flake defects is relatively low compared to the nodular defects. It mainly depends on the thickness and adhesion of coating on substrate fixtures and shields. To reduce the

density of flake defects all vacuum components should be cleaned after several batches. Although the concentration of flake defects is typically low, they can be very detrimental for the performance of the coating if they extend down to the substrate and expose it to the detrimental influence of the surrounding atmosphere (oxidation, corrosion) directly.

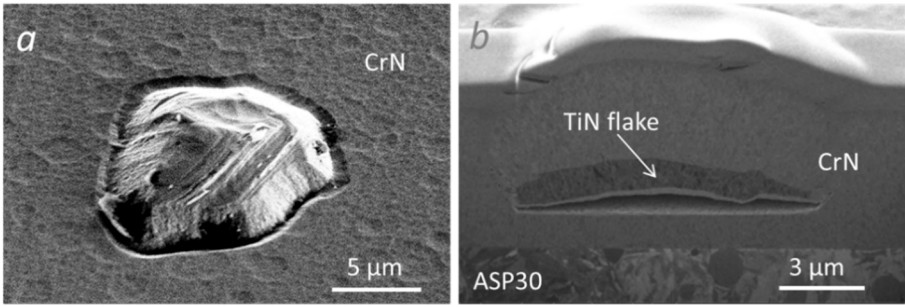

**Figure 14.** Top view SEM image (**a**) and FIB cross-section image (**b**) of flake defects in CrN hard coating prepared by evaporation using thermionic arc.

### 3.1.3. Droplet Defects

Droplet related defects are very common in the coatings deposited by cathodic arc deposition technique (Figure 15a). If metal ions are used for substrate etching, then droplets generated by arc discharge contaminate the substrate surface, which cause the formation of growth defects. While the generation of droplets continues also during deposition process the growth defects start to grow within the coating. As we will explain in Section 4.2.3 the metal droplets are produced due to the melting of the target. The liquid target material is ejected from the target as droplets. Part of these droplets can arrive on the substrate surface where the liquid material solidifies. Such droplets initiate defects in the depositing coatings.

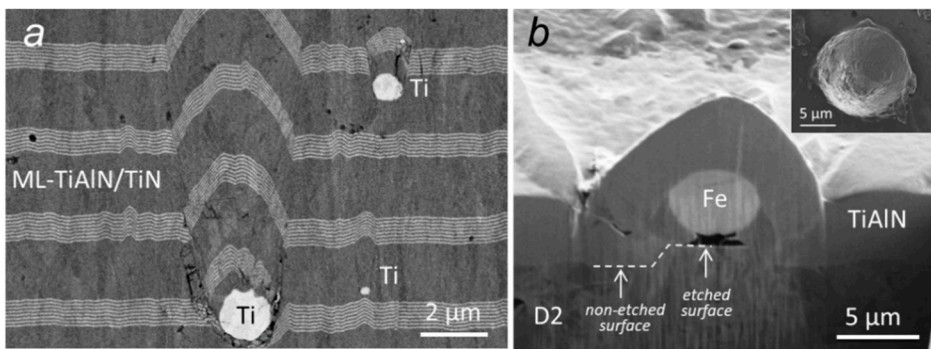

**Figure 15.** FIB cross-section image of a buried droplet formed in the multilayer TiAlN/TiN hard coating prepared at the company KCS Europe from Germany by cathode arc deposition technique (**a**). Fe-based droplet built in the TiAlN hard coating sputter-deposited on D2 tool steel substrate (**b**). The distinct step beneath the Fe droplet proves that it arrived on the surface of the substrate at the beginning of the ion etching.

We found that average height and average surface area of such type of growth defects are smaller in comparison with growth defects based on other types of seed particles. However, the number of droplets is more than 10-times greater. Droplets have a spherical or oval shape; therefore, the droplet related defects are of more regular shapes.

Droplets can be formed also during sputter deposition (Figure 15b), but on a much smaller scale. Their formation is caused by arcing on the substrate table and other inner components of the vacuum chamber. Therefore, composition of droplets is based mostly on iron.

## 3.2. Hole-Like Defects

In the literature, a great variety of terms are used in regard to the growth defects that are below the surface of the thin film. We will use the general term hole to describe any type of growth defect that is below the mean surface level of the thin film. Like protrusion defects, the hole-like defects can be distinguished with respect to their origin and shape. Figure 16 shows schematic classification of hole-like defects by their typical shape.

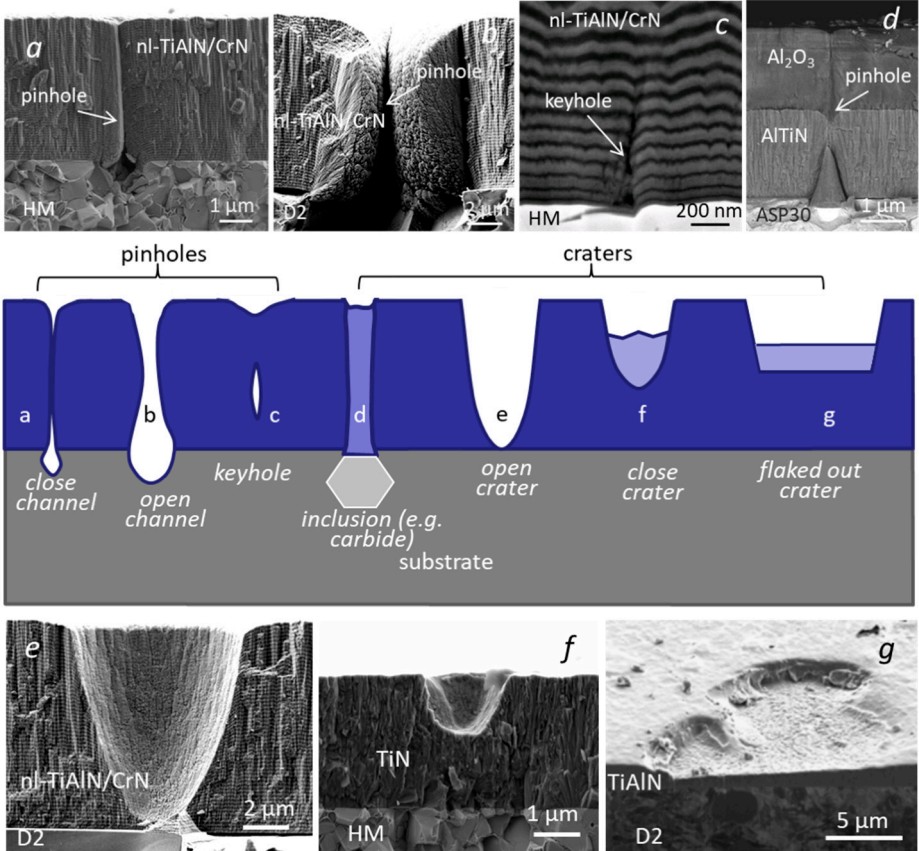

**Figure 16.** Schematic overview of different shapes of hole-like defects in PVD coatings: (**a**) pinhole at the site of a pit in the substrate; (**b**) open pinhole; (**c**) closed pinhole (keyhole); (**d**) pinhole that appeared at the site of a shallow crater formed during ion etching of ASP30 tool steel substrate; (**e**) crater formed due to the expulsion of a nodular defect; (**f**) a crater that does not extend through the entire coating; and (**g**) crater of irregular shape formed by detachment of flake defects. FIB and fracture cross-sectional SEM images of typical hole-like defects in PVD hard coatings prepared by magnetron sputtering (coating types: nl-TiAlN/CrN, CrN, TiAlN) and by evaporation using thermionic arc (coating type: TiN) are added.

### 3.2.1. Pinhole Defects

Pinholes as one of the most common growth defects in PVD thin films are discontinuities in the coating microstructure in the form of thin holes having a (sub)micron size diameter and extending from the substrate to the top surface of the coating. There are a number of causes for formation of pinholes (Figure 16). A majority of pinholes are generated at the substrate imperfections, such as cavities (pits) or shallow depressions formed on the substrate surface during its pretreatment. The usual origin of pinhole formation is geometrical: a narrow but deep cavity, where the shading effect prevents the film growth on the cavity walls. Namely, the coating is preferentially deposited on the flat front side of substrate, while the deposition rate on the sidewall of the cavity is much lower. Due to the shadowing effect, coating on the sidewalls of the cavity has a columnar, porous structure.

The angular distribution of the impinging vapor flux on the surface is the most important factor which influences the size and the number of pinholes generated by geometrical shadowing [53]. The more random the flux direction is, the smaller the number and size of the pinholes (Figure 17a,d). A random vapor flux direction is established: (i) by using substrate-holding fixtures that randomize the substrate position and angle of incidence in the vapor flux and (ii) by using extended vapor sources or several vapor sources. As compared to evaporation, conventional sputtering can provide more conformal coatings over protrusions and low aspect ratio cavities. This is because sputtering sources form a broad atom flux and atoms are ejected at wide angles too.

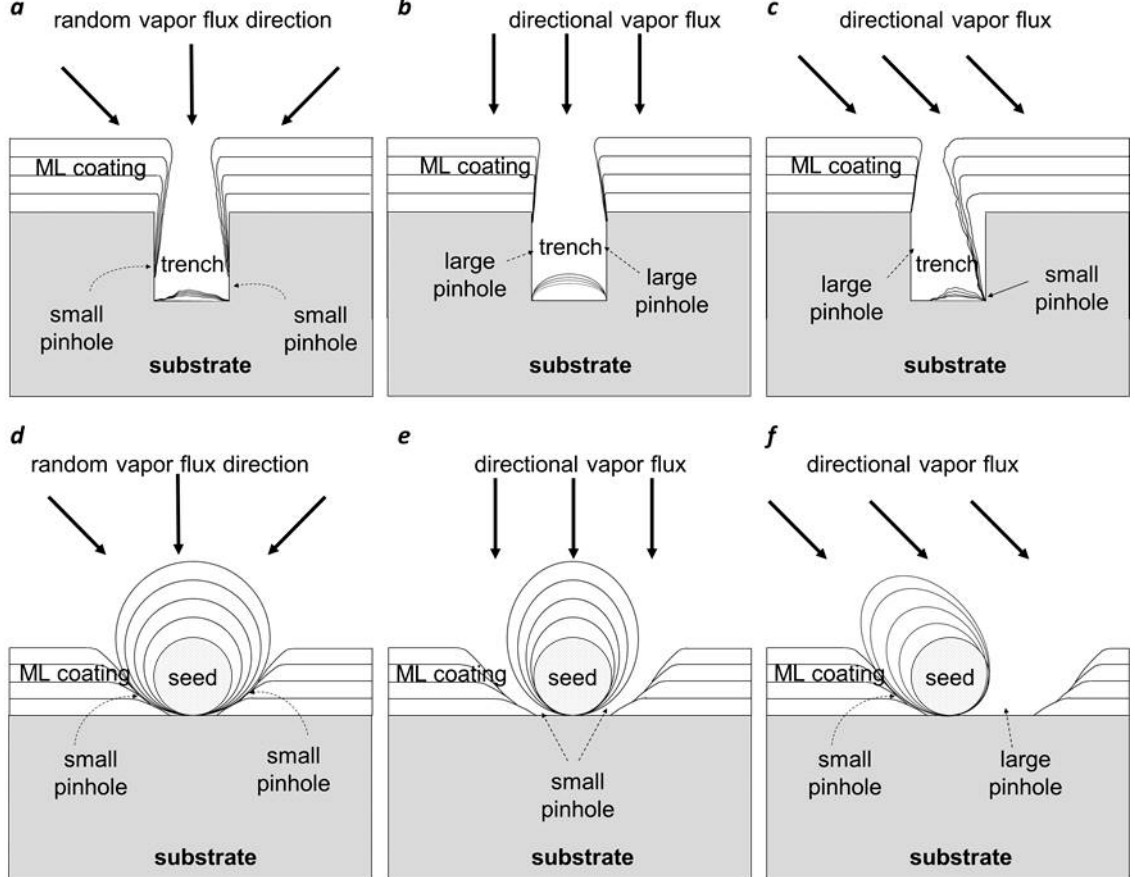

**Figure 17.** The schemes show the distribution of coating around two kinds of surface features (high aspect ratio trench, sphere) and formation of pinholes for a range of angles of incidence of depositing flux of atoms.

Whether a pinhole will be formed or not depends on the aspect ratio of the hole (depth/hole diameter) and not its size. In the case of a narrow but deep hole (high aspect ratio) on the substrate surface the shading effect prevents the coating growth on the hole walls. When the aspect ratio is high, the deposition starts to coat the upper sidewalls and corner of the feature, which shadows the lower area from subsequent deposition. During the following coating growth, the high aspect ratio crater is narrowing. If the PVD film is thick enough they even appear to touch and close the opening, forming an isolated pore, called a keyhole. However, a microstructural discontinuity is preserved through the growing coating, extending up to the coating surface. On the other side a laterally large but shallow hole (low aspect ratio) will not develop a pinhole. In this case an open pinhole will form. What will happen to the high aspect ratio holes on the substrate surface depends also on the thickness of the coating. If the width of the crater on the substrate surface is comparable to the final coating thickness, the pinhole will not be able to close up during the coating growth.

The pinholes are also found everywhere where there are nodular defects, which start to grow at the substrate surface. As already mentioned, the contact between the nodular defect and the undisturbed coating is poor. The contact of the seed particle with the substrate is very poor too as there is no coating at all. Therefore, the border between the nodular defect and the undisturbed coating is essentially a "circular" pinhole.

Through-porosity (pinholes) can be detected by using SEM, selective substrate liquid chemical etching, selective plasma etching, electrolytic copper decoration, or it may be measured by corrosion potentials (anodic polarization) [54].

Typically, pinholes occupy a relatively small area of the total coated surface. The significance of pinholes is highly dependent on the application. In some, pinholes are functionally insignificant, whereas in other cases, they are intolerable. Their influence on the functional properties of thin films is discussed in Section 5.

### 3.2.2. Crater-Like Defects

As discussed above, the bond between the nodular defect and the surrounding matrix as well as between the seed particles and the substrate is poor. Due to the buildup stresses in the growing coating, some of the nodular defects detach from the coating, leaving a crater on the coating surface. The resulting crater can be interpreted as "inverse" nodular or flake defects. Formation of such a crater may also be caused by external forces such as cleaning by ultrasonic cavitation or wiping after the coated sample is removed from the deposition chamber. Depending on the moment of the spall-off (during or after the deposition), areas of bare substrate may be found at the bottom of these holes. If the nodular defect leaves the coating during the deposition process, then the created crater is covered by the still growing coating. The microstructure of the overgrowing coating is highly columnar and porous with poor cohesion. Figure 18a shows a fracture cross-sectional SEM image of a crater-like defect formed from the nodular defect after the deposition was completed. The nodular defects probably detached during the cooling stage, where the internal stress was augmented by the thermal stress.

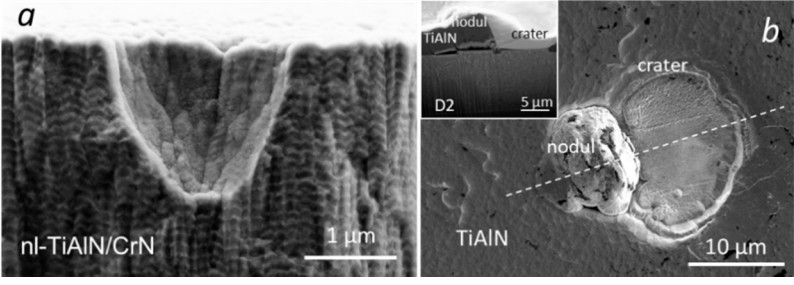

**Figure 18.** Crater-like defect left by the detachment of a nodule in sputter-deposited nl-TiAlN/CrN hard coating (**a**) and combination of flake and nodular defects in sputter-deposited TiAlN coating (**b**).

Similar to craters formed from the nodular defects, large craters of irregular shape are formed by detachment of large flake defects when the internal stress overcomes the adhesion (Figure 18b). In the case when this happens during the coating deposition, the remaining crater with the flat bottom is covered by the additional growing coating. The concentration of such defects is not large, but they cause large imperfections in the morphology of the thin film and can have large detrimental effects on the properties of films. This is especially the case when the craters are extending down to the substrate.

### 4. Origin of Seed Particles

As mentioned in previous sections, any protrusion on the substrate surface is a seed for formation of nodular defects. In general, there are several possible seed origins [18,55] which are independent of the deposition techniques. The seeds for nodular defects can form from: (a) geometric irregularities

on the substrate surface after mechanical pretreatment (see Section 2.1); (b) substrate irregularities arising from ion etching (see Section 2.3); and (c) foreign particles that arrived on the substrate surface before or during the coating growth. In all these cases the consequent growth mechanisms are similar, yielding similar nodular defects, which are generally indistinguishable from a top view. The overall shape does not depend much on the seed type, nor its chemical composition.

### 4.1. Foreign Seed Particles

Different seed particles (e.g., dust, debris, polishing residue, impurities) remain on the substrate surface after its mechanical pretreatment, wet cleaning, drying, and batching. In order to obtain a smooth coating surface, we remove the majority of particles in the production process [55]. To approach this goal a high-quality wet cleaning procedure must be performed in an ultrasonic bath. First the steel substrates have to be demagnetized to avoid the problems in removal of ferrous debris from its surface during wet cleaning in alkaline cleaning agents. The use of ultrasonic cleaning in addition to manual cleaning has a positive impact on reduction of foreign particles. Manual scrubbing has been shown to be critical to reduce surface particulates. After rinsing in deionized water, the substrates must be dried in hot air as quickly as possible because the residual water film will stick dust particles to the surface. Fine air and water filters must be used in order to minimize the concentration of particles in deionized water and dry air. In order to minimize the re-contamination of the cleaned substrate surface before it is placed into the deposition chamber, we have to ensure a clean processing environment, a proper storage after the external cleaning, and adequate handling during batching. Seed particles can also be brought in the deposition system with fixture components. Inadequate substrate cleaning and/or inadequate cleanliness during transport and batching drastically increase the density of detrimental particles. Figure 19a, for example, shows a nodular defect that originates from a $CaCO_3$ seed particle. Such a particle is probably the residue of cleaning agent.

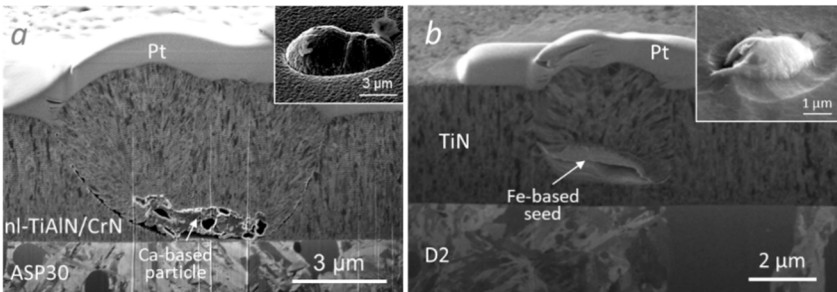

**Figure 19.** Top-view SEM (insets) and FIB images of nodular defects originating from Ca-based and Fe-based seeds. Nanolayer nl-TiAlN/CrN (**a**) and TiN (**b**) hard coatings were prepared by sputtering and evaporation using a thermionic arc, respectively.

An intensive generator of seed particles is the deposition system itself. Some particles may fall on the substrate surface already during rough pumping because of possible turbulent gas flow which may pick up small particles accumulated on the bottom of the vacuum chamber from previous batches. To avoid any turbulence in the gas flow, slow pump-down during rough pumping (up to 100 mbar) and "soft-venting" must be used.

The next origin of seeds is wear particles (debris) generated by all moving components in the vacuum chamber (opening and closing valves, moving parts of fixturing) (Figure 19). More motion elements in the fixture systems mean more wear particles. In particular, the problems are triggers that are often used for discontinuous rotation of substrates. Some debris also originate from maintenance and installation (e.g., wear of hand tools, insertion of bolts). Wear particles may be minimized by using appropriate non-galling materials in contact, vacuum-compatible dry lubrication of surfaces in contact, smooth surfaces and minimal contacting forces. Upward-facing samples have a higher defect density

than downward- or lateral-facing samples because of particles falling on the substrate surface due to gravity. Therefore, mounting substrate surfaces facing upward should be avoided.

The seed particles may also be brought in the deposition system with processing gases. The use of high purity gasses is strongly recommended.

After several deposition runs the coating buildup on the shields and fixtures becomes too thick, and it may flake, particularly if high residual and thermal stresses are present in the coating material. The flakes, which are transferred through plasma, build up a negative charge. They are held to the substrate surface by electrostatic forces, which affect the micrometer-sized particles much stronger than the gravitational forces. One way to reduce this problem is to occasionally overcoat the brittle and poor adhered deposit of hard coatings with a softer (pure metal) material; this process is called metal layer pasting [56]. Pasting is a high-power sputtering step in metal modes that cleans up the sputter surface and also seals the re-deposited nodules with thin metal layers. If pasting is not done at recommended intervals of time, flaking of re-deposited materials from the sputtering target, shields, and fixturing is likely to cause particle formation in the deposited coatings. In any case, regular sand blasting of the deposition chamber components (e.g., fixtures, shields) is necessary to perform.

Another possible origin of dust particles is their formation in plasma [57]. Researchers in the semiconductor industry realized that sub-micrometer particles can be formed in chemically reactive plasmas (e.g., in plasma-enhanced chemical vapor deposition processes (PECVD)) by the gas-phase reaction and aggregation of atoms or molecules from etching or sputtering processes. The possibility of particle formation is smaller if the partial pressure of the reactive gas is reduced.

As mentioned above, many sources of seed particles can be eliminated using proper vacuum and substrate handling techniques. In the case of providing the cleanest conditions before coating deposition, the main source of seed particles is then the source material itself (evaporation crucible, cathodic arc or sputtering target). Each deposition technique and material combination is unique and must be studied individually. In the following text, the origin of seed particles for these three different evaporation sources is discussed in more detail.

### 4.2. Seed Particles Originating from Deposition Sources

### 4.2.1. Seeds in Electron Beam Evaporation

In addition to above described general origin of seed particles (Figure 20), specific seed particles are characteristic for electron beam evaporation. Various mechanisms of seed ejection from an electron-beam process are possible. Heating of the source material can produce seeds by several mechanisms including: (a) explosions caused by the heating of gas inclusions or micro-arcing; (b) splashing of molten material; (c) electrostatic repulsion of charged particles; (d) thermal-induced cracking; and (e) temperature-induced solid-state phase transitions [18,53].

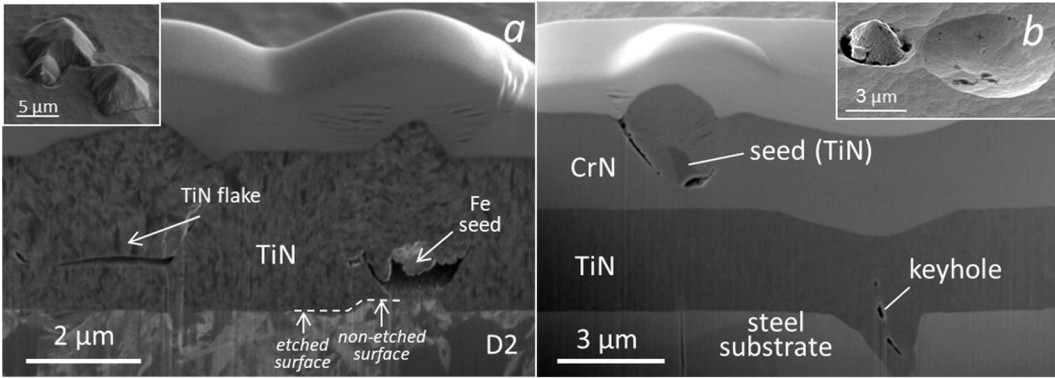

**Figure 20.** Typical growth defects in TiN (**a**) and TiN/CrN double layer (**b**) hard coatings prepared by thermionic arc evaporation. FIB images show the origin of defects.

In the deposition from molten source material, small droplets may be ejected from the molten pool, which lands on the substrate and is incorporated into the coating during the deposition process. This, so-called spitting phenomenon can be caused by the release of gas or vapors in the molten material during rapid heating. The source of spits can be suppressed by using pure evaporation material, preheating for degassing the evaporation material or slow heating to vaporization. In addition to these measures, the selection of deposition parameters is also very important. In the case of e-beam evaporation the key deposition parameters are e-gun voltage, e-gun emission current, and beam pattern. A low current allows the source to operate at relatively low temperatures where effects due to charging, stress relief, and phase transitions are reduced. Setting the e-gun at lower accelerating voltage keeps the e-beam heating nearer to the surface of the melt (about 25 μm in depth), thereby minimizing thermal gradients. A broad and rapidly scanning electron beam decreases the power density into the source. The splitting occurs if e-beam energy is delivered at a rate faster than the coating material can accommodate this energy by evaporation, conduction, or radiation.

### 4.2.2. Seeds in Magnetron Sputtering

Magnetron sputtering is one of more commonly used methods for the fabrication of PVD thin films. In magnetron sputtering, operated either in DC, RF, or pulsed mode, the plasma is generated over a large area of the cathode as opposed to the cathodic arc where plasma is localized in a small area of the arc. In the sputtering, material from the target is vaporized as individual atoms or groups of atoms and therefore, in principle, does not generate micro-droplets. In the case of magnetron sputtering the formation of specific seed particles (flakes) are caused by: (a) flaking of cones formed in the target racetrack, (b) flaking of the redeposited nodules from the target surface, and (c) by arcing [56]. In the following text, all three mechanisms are discussed in more detail.

### Formation of Cones on Target Surface During Ion Bombardment

Different physical phenomena (e.g., cone formation, faceting, trenching) occur at the target surface during ion bombardment and cause its roughening. Here only cone formation as an extreme case of surface roughness is discussed. The formation of cones or micro-protrusions on target during ion sputtering in the presence of a seed material was first observed several decades ago by Wehner and Hajicek [58]. They found that cones formed by sputtering of the Cu target surface with a concurrent supply of impurity Mo ions. In this case the Mo atoms create regions with lower sputtering rates which cause a local masking effect. The substrate material around the impurity center is sputtered away faster because of a much higher sputtering yield.

Apart from these local masking effects, cones can also develop if some degree of impurity atoms is present in the target material. For example, if aluminum targets of the purity 98% and 99.99% are sputtered under identical conditions only the target of lower purity shows the cone-like features formed on its surface. The formation process of conical protrusion can be explained by considering not only the primary erosion process (i.e., different sputtering rates, the variation of sputtering yield with the ion incidence angle), but also by other effects such as ion reflection, re-deposition, and surface diffusion. The undesirable effect of cone formation is the breaking and formation of seed particles that could be built into the growing coating. The distance between the target and substrate is rather small (50–120 mm); therefore, any particle generated during sputtering has a high probability to reach the substrate surface due to electrostatic repulsion.

### Flaking of Re-Deposited Nodules from the Target Surface

During magnetron sputtering in a reactive mode (oxides or nitrides) re-deposition of sputtered material occurs from the racetrack to the center and on the edges of the planar magnetron. In this region the plasma density is low and sputtering does not take place. Therefore, the material is deposited on the perimeter rather than eroded. Due to the internal stresses the redeposited material grows in the form of filaments [59,60] (Figure 21). During deposition the filaments gradually grow. Those which

form near the racetrack cross the high-density regions of plasma and are resistivity heated (due to increased current flow). Heating of the filaments causes its fracture and ejection of the fragments. The ejected fragments become electrically isolated from the target and charge negatively. The charged particles are accelerated away from the target due to the repulsion effect [58,61]. Some of them arrive to the substrate surface and become seeds for the growth of nodular defects in the coating. Such a mechanism for particle formation was first observed in carbon targets used to deposit diamond-like coatings on magnetic disks and during nonreactive sputtering of TiN [60].

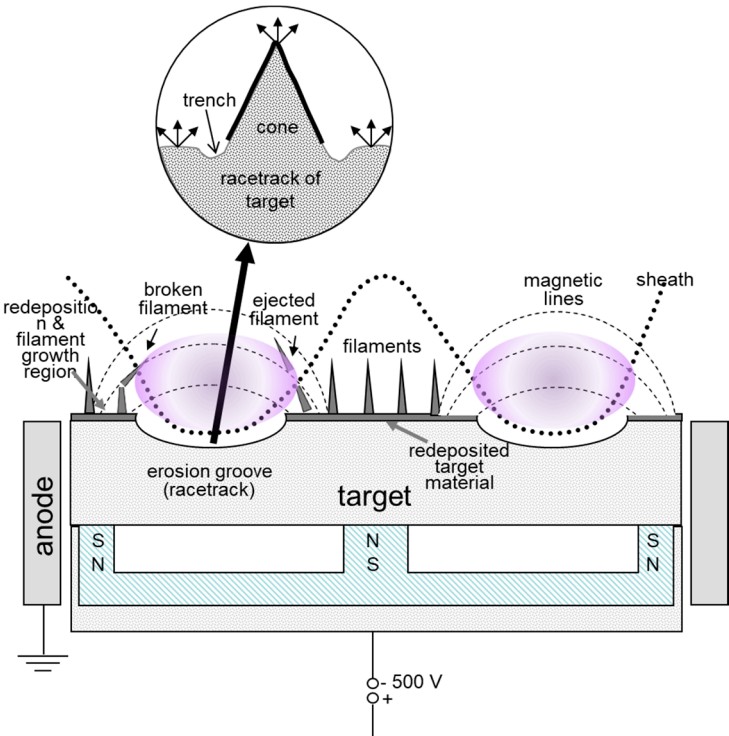

**Figure 21.** Scheme illustrates the formation and fracture of filaments on the perimeter and in the middle of the magnetron target surface. The cone formation on the racetrack of target is also schematically shown.

Filaments on the target surface present an important source of contamination in the sputtering process. The formation of filaments can be avoided if a cylindrical magnetron is used. On the cylindrical target a re-deposition zone does not grow, and this is the reason why such magnetron sources provide much higher process stability and cleaner coatings can be produced.

In order to prove that parts of seed particles really originate from the target surface, we sputtered deposited TiAlN coating on the test sample in a production batch together with different hard metal and high speed steel (HSS) cutting tools. After deposition we analyzed the broken nodular defects found on the coating surface by backscattered-electron (BSE) imaging (Figure 22a,b). The BSE image reveals the internal structure of nodular defects, which includes the microstructure of the coating as well as the size and shape of the seed particle. Additionally, a bright layer around the seed particles was observed. EDX analysis confirms that the seed is a TiAlN flake, while the bright layer (app. 50 nm thick) is composed of iron, tungsten, and chromium. The presence of these elements can be explained by contamination of the target surface during ion etching. As mentioned earlier, a lot of weakly bonded seed particles are present on the target surface, especially outside of the racetrack. During ion etching these particles are covered by the thin film of batching material. Immediately after starting the deposition an electrical charge may accumulate on these particles and electrostatic forces cause their self-expulsion. Some of them can be built in the coating growing on the substrate surface.

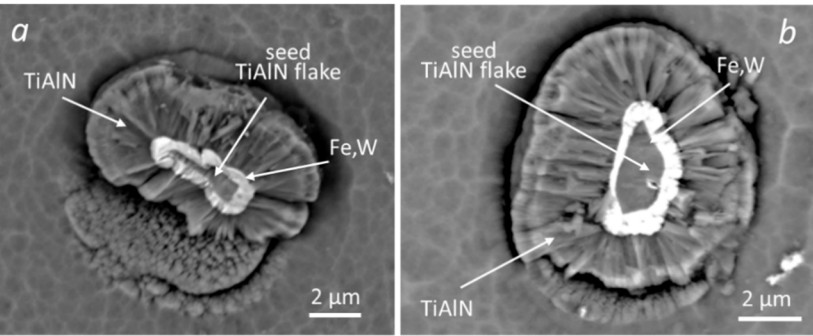

**Figure 22.** SEM images of the broken nodular defects in the sputter-deposited TiAlN hard coating. Seed particles are clearly visible at the fracture. The origin of these seed particles is the target of magnetron sources.

From a large number of FIB cross-sections of individual nodular defects, we found that the majority of defects in the magnetron sputtering originate from the seeds on the substrate surfaces. This finding supports the above-mentioned statement that a part of the seed particles arrives on the substrate surface just before or at the start of the deposition process.

Arcing

Arcing is a common problem in both metal and reactive magnetron sputtering because it is a significant cause of defect generation and process instabilities [62–64]. An arc appears due to the rapid accumulation of charge on a small area (e.g., nonmetallic inclusions) during the ion bombardment of the target (Figure 23). If the local electric field exceeds the dielectric strength of the insulator an electrical breakdown occurs. In the case of reactive sputtering of oxides, an isolating (or less conductive) layer may grow in the transition zone from the racetrack to the non-sputtered region, leading to the tendency of micro-arcing and thus generation of droplets (macroparticles). Native oxide and nonmetallic inclusions on the target surface are not the only cause for arcing. A similar effect occurs when the target material (e.g., powder metallurgy targets) is not fully densified or if microvoids, produced during target manufacturing are present in the target material. Pores and microvoids can trap gases, which are released during target ion etching. Locally high pressure regions of gases cause arcing and thus generation of particles.

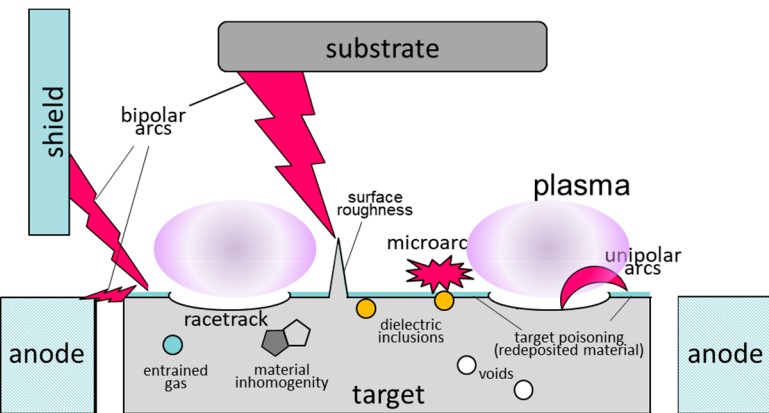

**Figure 23.** The scheme shows imperfections of metallurgical origin in the target material that may cause three arcing modes (unipolar arc, bipolar arc, and microarc).

There are several ways to reduce arcing on the targets. The first approach is conditioning of the targets after each deposition run. Conditioning means that the target power should be ramped up slowly in order to eliminate any native oxide or other surface contamination. Arcing can also be

prevented or suppressed by using advanced power supply units. Novel power supplies are equipped with a sophisticated arc detection system which is able to detect arcs much faster (detection time is less than a microsecond), thus the amount of energy released to an arc is much smaller. The next approach is the use of the pulsed sputtering technique, developed in the mid-1990s. In this case the charge built up by ion bombardment can be compensated by electrons which arrive at the target surface during the positive half cycle of the pulse.

The problem is not only the arcing on the target but also arcing on the substrate table and other components of the vacuum chamber. Thus, the arcing on the substrate table and shields during the ion etching and coating deposition steps additionally stimulate the flaking phenomena. Due to the high energy which arises during arcing at certain points of the substrate table, anode, and other components of the vacuum chamber, locally high thermal stresses appear. The fragments of the growing coating delaminate from a spot somewhere in the deposition chamber and fall on the surface where they can get stuck. The fragment may even originate from a previous deposition of the same coating type. In general, they have the same chemical composition as the undisturbed coating. Therefore, it is practically impossible to distinguish such types of seed particles from the current growing coating in the cross-sectional fracture SEM image. Although in multilayer coatings they can be clearly visible from the shape of layer contours (Figure 24).

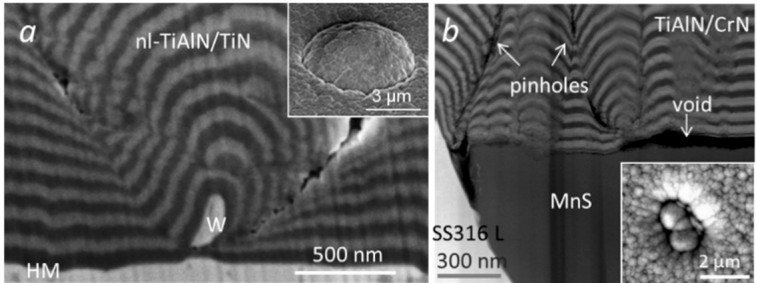

**Figure 24.** The seed particle in the nodular defect (**a**) or pinholes (**b**) can be clearly identified from the shape of layer contours in sputter-deposited nl-TiAlN/CrN hard coatings.

In hard coatings prepared in a sputter deposition system equipped by hollow cathodes used to assure more intensive plasma (plasma booster technology) during the etching step, we also observed rather large nodular defects with a typical diameter of about 20 μm (Figure 25a). We found that they started to grow on the substrate surface on the seeds composed of copper and tantalum. The origin of both elements is arced between the tantalum tube (nozzle) and the copper anode in the hollow cathode source, which appears occasionally. From the nodule shape we can conclude that the copper component of droplets settled on the substrate in the liquid state, while the tantalum component droplets were in the solid state. Beneath the seed (droplet) a step on the substrate surface is visible. It was formed during the ion etching process, because the droplet shaded the substrate surface. This step confirms that the Cu-Ta fragments arrived on the substrate surface during the early stage of the etching process. On the other hand, there is no step beneath the seed particle if it arrived on the substrate surface at the beginning of the deposition process.

We should also mention the ion beam sputter deposition, which is commonly used in the production of high-quality optical coatings [64]. This technique produces the lowest defect density among all PVD techniques. In this technique, target material is sputtered by an external ion source. The target is not on any electric potential; therefore, it cannot produce arcs.

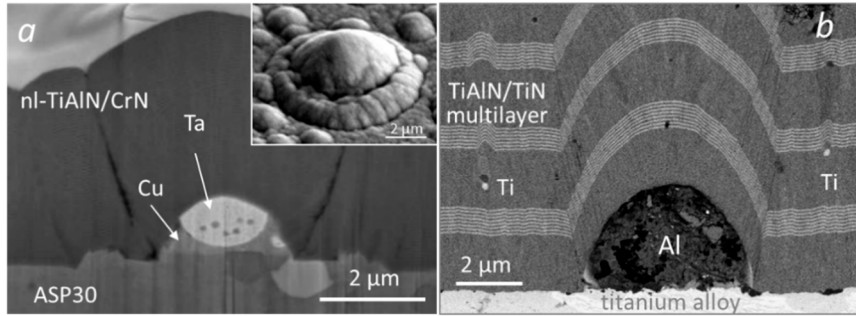

**Figure 25.** FIB image of a droplet-related defect in sputter-deposited nl-TiAlN/CrN hard coating. The seed composed of cooper and tantalum was formed during intensive (booster) ion etching (**a**). Two small Ti buried droplets and one large Al-based hemisphere with a flattened bottom formed during preparation of TilN/TiN multilayer using the cathode arc deposition technique (**b**). The multilayer TiAlN/TiN hard coating on the titanium alloy substrate was prepared at the company KCS Europe from Germany.

### 4.2.3. Seeds in Cathodic Arc Evaporation

In cathodic arc evaporation the majority of growth defects originates during the coating deposition [65,66]. The cathodic arc discharge between an anode and a cathode is localized in small spots (typically only a few micrometers in diameter) and presents an intense source of plasma with a current density of about $10^6$–$10^{12}$ A/m$^2$. The spot, which moves over the cathode surface, causes local melting and evaporation of the target material (cathode). The result of the arc evaporation process is not only ions and atoms from the cathode surface but also significantly larger particles (droplets). Formation of such droplets is a result of plasma pressure on the liquid cathode material. The majority of droplets have a diameter less than 1 µm. The droplets preferentially splash under a relatively shallow angle with respect to the cathode surface and since they originate during the coating deposition. they can be found at all coating depths.

On the way away from the cathode the droplets cool down. Small droplets (less than 1 µm in diameter) cool more rapidly than coarse ones. It has been shown that small droplets settle on the substrate in the solid state, while the coarse ones settle in the liquid state. Therefore, smaller droplets have a nearly spherical shape (Figures 15a and 26a,b), while the larger ones are less regular in shape (flattened droplets, Figures 15b and 25a,b). Namely those droplets which reach the substrate surface in the liquid phase, likely change the geometry upon impact. Touching the surface, these types of droplets will be deformed and quenched. Once they arrive on the surface they may stick and be incorporated into the growing film. Those droplets that become solid before impact have a high probability to reflect. While they reach the film continuously, some may be overgrown. The fact that the droplets are found at different distances from the substrate-coating interface reveals that they are generated and incorporated within the coating during the entire coating deposition process. The embedding of droplets causes the formation of nodular defects on the growing film.

The size and amount of the droplets are primarily affected by the cathode material. In general, materials with a high melting point generate less and smaller droplets. The other deposition parameters (e.g., deposition temperature, gas pressure, arc current, and power) also affect the formation of droplets. For example, increasing the reactive gas pressure in the chamber results in the generation of fewer droplets, which is due to the formation of compounds on the cathode surface. The compounds usually exhibit a higher melting point than the original target. The formation of intermetallic phases and high-melting point thin ceramic layers on top of compound cathodes may strongly influence the behavior and movement of the cathode spot and therefore the droplet generation. For instance, an increased cathode temperature leads to a higher number of generated droplets due to the larger area of the molten target material. The cathode spot movement can be used to decrease the average

cathode temperature based on the reduced average arc spot rest time, equivalent to a decreased local thermal impact.

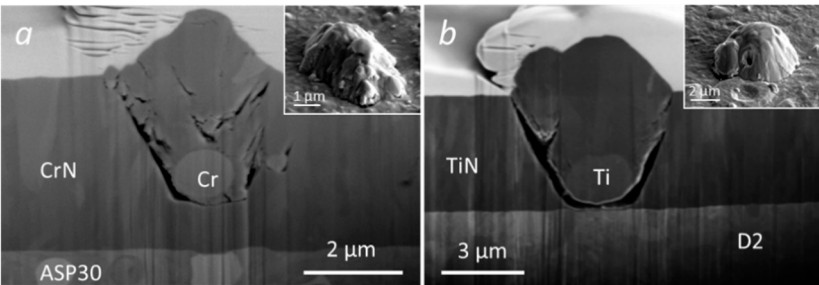

**Figure 26.** Typical droplet-related defects in CrN (**a**) and TiN (**b**) hard coatings prepared by cathodic arc evaporation.

The detrimental effects of droplets include local loss of coating adhesion, surface roughening, grain coarsening, nonuniform phase, and composition within the droplet [67]. The presence of droplets also leads to the creation of pores because some of the droplets are wrenched out due to high compressive stresses.

There has been a lot of research on how to minimize droplet formation or droplet incorporation into the growing film. Numerous approaches have been proposed. The simplest means to reduce the number of droplets in the vacuum arc coating synthesis are as follows [68,69]:

- increasing the arc speed on the cathode surface by using a magnetic field; in this way, the arcs are moving faster on the cathode surface, therefore they melt a smaller volume of material;
- reduction of the temperature of the cathode surface by intensive cooling;
- reduction of the arc current in order to reduce the density of ion flow;
- low-angle shielding of cathode; the majority of the droplets are emitted at angles lower than 30° with respect to the target surface;
- droplet filtering involves guiding the plasma towards the substrate using an electromagnetic field (0.01–0.1 T); in contrast to electrons and ions, the droplets are not charged and therefore will not follow the non-linear path to the substrate;
- the use of higher partial pressure of the reactive gas during deposition due to the formation of compound layers with a high melting point;
- the number of droplets can be reduced with increasing bias voltage; the latter may be attributed to the effect of the enhanced ion (re)sputtering and deflection of the negatively charged droplets.

## 5. The Influence of Growth Defects on Functional Properties of Thin Films

In this chapter we provide an overview on the research related to the influence of the growth defects on the functional properties of thin films and coatings. We start with the role of growth defect on the optical properties and in the semiconductor devices, which were historically studied first. Then we discuss the influence of defects on wear and friction of coatings, corrosion and oxidation resistance, surface wettability, and permeability of gas barrier coatings.

### 5.1. Optical Properties

Thin films for optical applications have been produced for many years and today they are an integral part of the majority of modern optical systems such as lasers, displays, lighting, mirrors, anti-reflection coatings, beam splitters and filters, decorative coating, security (antiforgery) devices, and others. Optical coatings have been traditionally deposited by evaporation (from either a crucible or an e-beam source) and sputtering, frequently assisted by ion bombardment. Optical coatings often present a critical part for the entire optical system.

The limiting aspect of optical coatings is especially their failure to perform high-power laser illumination. Several studies demonstrated that the low damage threshold of high-power laser optical components is associated with nodular defects in the optical coating. Due to the lens-like shape the nodular defects focus the light within the defect and thus light intensification is significantly greater than that in the defect-free regions. Therefore, it leads to local overheating and consequently to coating damage (crater-like pits). A few localized defects in optical interference mirror coatings can substantially increase the scattering loss to tens or hundreds of ppm [27].

Bercegol [70] showed that laser conditioning of some optical components (dielectric multilayer coatings) for high-power lasers can improve the functional laser damage resistance. This method is based on an under-threshold pre-radiation of the coating by laser, which causes a gentle ejection of nodular defects. Smooth-edged pits are left behind.

Another promising approach to reduce the density of nodular defects was proposed by Mirkarimi et al. [71,72]. Their proposal is based on integration of the thin film deposition process and direct etching of the film/substrate at normal incidence. The process consists of 50 nm silica deposition followed by ion beam etching of one half of the deposited layer. When etching a nodular defect, the sides of the defect etch faster than the top because the etching rate at normal incidence is much smaller than at high incident angles (~50°). The enhanced etching at the sides can cause the nodular defect to shrink until the defect gradually disappears (Figure 27). The geometric minimization of the coating defect significantly improves the laser resistance of the optical coating. Unfortunately, this process is not effective for the planarization of pits and scratches.

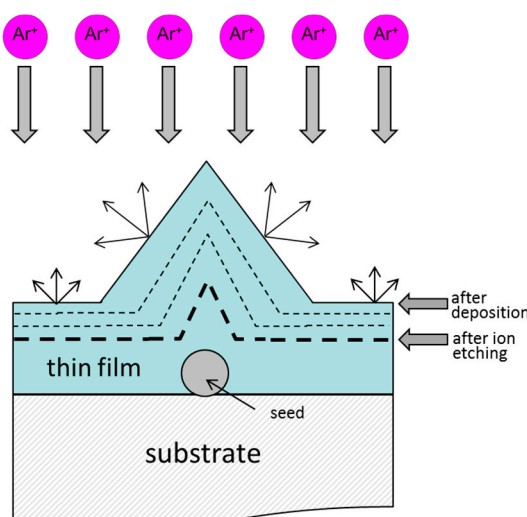

**Figure 27.** Illustration of the ion etching process applied for planarization of the coating surface.

*5.2. Growth Defects in Semiconductor Devices*

In semiconductor thin film device manufacturing (e.g., integrated circuits, flat panel displays, magnetic and optical storage media, photovoltaic devices, and other thin film devices), particle contamination and consequently growth defects are a serious problem because they cause a decrease in production yield and reliability problems during product use [29]. On magnetic storage discs, particle contamination can result in read-write errors, bad sectors, and total disc failure. In the manufacturing of complementary metal-oxide-semiconductor (CMOS) integrated circuits, particle contamination can cause pinhole formation, delamination, and interconnection shorts or opens in the metallization processes. Therefore, it is important to understand the formation of localized defects in thin films which compose the semiconductor device. In principle every process step in production of such devices (film deposition, lithography, etching) could be a source of contamination. The control of particle contamination is especially of great importance in advanced microelectronic technologies such as

extreme ultraviolet lithography where defect-free masks are necessary [72]. To control these deleterious effects during semiconductor fabrication, great care is required to reduce surface contamination levels during the handling and manufacture of devices [73]. Elimination of particles that cause the so called "killer defects" during thin film processing is therefore one of the biggest challenges for semiconductor device producers.

### 5.3. Friction and Wear

In tribological applications the roughness of a coated component has an important influence on the friction, wear, and tendency for a material to be transferred [74]. Protruding nodular defects result in high abrasive wear of the counter material and due to the micro-ploughing and material pick-up effects, the friction coefficient increases. As long as protruding nodular defects are present on the coating surface the friction coefficient does not reach the steady state. The rougher the sample, the longer the time required to obtain the steady-state value of the friction. The period until a conformal sliding contact is formed is called the running-in phase. During the operation of tools protected by PVD hard coatings the most common reasons for their failure are coating fracture, coating delamination, and subsurface fracture (substrate failure) [75]. Such failures can be caused by cracks that initiate at different coating defects. The first contact between the tool and the workpiece material, which move relative to each other, always takes place at the highest peaks of the surfaces [76]. In addition to substrate surface asperities, such contact spots are various coating imperfections (e.g., droplets, nodular defects). Due to the small actual contact area (about 10 % of the surface area) the contact pressure at these spots is very high [77]. High pressure and shear stresses at the nodular defects cause the formation of cracks and therefore they collapse into small fragments of hard coating material. The second mechanism of wear particle generation and friction increase is caused by interlocking and breaking of all types of asperities. It all means that the nodular defects are the primary source of abrasive particles in the sliding contact.

Based on this description we can understand why wear is so intensive during the early stage, rather than later on. During this process the protruding surface asperities in the PVD hard coating and wear particles abrade the surface of the softer counter-body material. Such a plowing effect causes a high wear rate of the counter surface. During further sliding the asperities are gradually removed and both surfaces are fitted together. Consequently, the contact area increases. As soon as a smooth surface is formed the contact pressure is reduced and consequently the risk of fatigue damage is reduced. During this period the transition from the mechanical wear-dominated friction to the adhesion-dominated friction takes place [77]. Additionally, any topographical imperfections on the coated forming tool surface also present a potential initial point for the onset of galling and transfer of the workpiece material [78–80].

Only a few papers can be found in the literature that paid attention to how growth defects influence the tribological properties of PVD hard coatings. Poulingue et al. [24] designed an experiment to analyze the damage initiating from nodules through a purely mechanical approach. The artificial nodules were generated by dispersing diamond seed particles on the polished aluminum substrate before deposition. Mechanical damage was progressively induced by pulling the samples in tension in an SEM. In-situ observation showed that cracks first arise at larger nodular defects. Fallqvist et al. [40] showed that in a sliding contact, growth defects have a strong impact on the tribological behavior of the coating causing abrasive wear of the less hard counter material surface and material transfer to the coating. Both mechanisms affect friction characteristics. In order to reduce the amount of surface irregularities introduced during the coating process, they recommended post-coating polishing. In such a way it is possible to reduce the material-transfer tendencies and to stabilize the coefficient of friction in the sliding contact. Luo [39] studied the running-in period in magnetron-sputtered TiAlN/VN multilayered coatings against an alumina counter body in a large temperature range up to 700 °C. He found that during the running-in period the growth defects collapse, while the fragments are released into the wear track and form wear debris. The effect of arc-evaporated droplets on the

wear behavior was investigated by Tkadletz et al. [41]. In their work special emphasis was given on the role of droplets in the performed ball-on-disk tests, where possible mechanisms triggering coating degradation were determined. Recently, we studied the influence of the surrounding atmosphere [81] and nodular defects on tribological behavior of sputter-deposited TiAlN hard coating using a new method. The novelty of our approach is based on cycle-to-cycle experiments [42,43]. We pinpointed selected defects on the coating surface and then followed them through the tribological test with a scanning electron microscope (SEM) and a focused ion beam (FIB) microscope. This approach gave us insight into the processes occurring after a certain number of sliding passes (Figure 28). The tribological tests we performed highlighted the importance not only of the nodular defects, but also of the protrusions that are located at carbide sites in different tool steels [43].

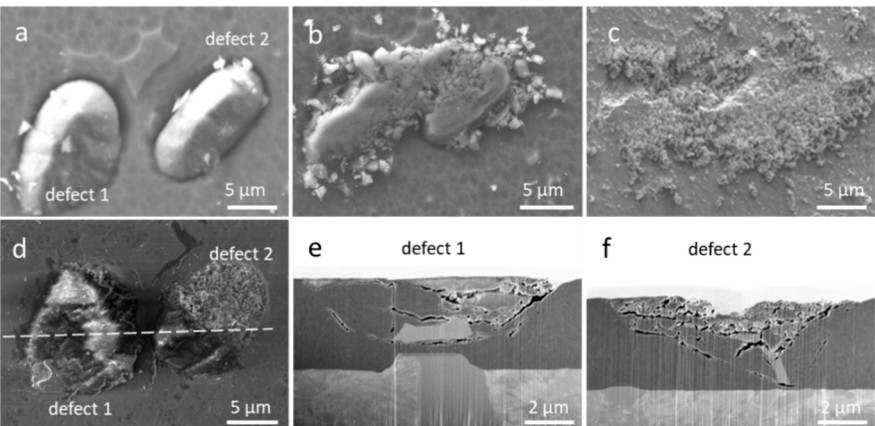

**Figure 28.** SEM images of the same nodular defects in the sputter-deposited TiAlN hard coating: as-deposited (**a**) and in the wear track after 1 cycle (**b**), 4 cycles (**c**), and 128 cycles (**d**) using an alumina ball (2 N, 2 Hz). FIB cross-section images (in direction marked with the dashed line) of both defects after 128 cycles (**e**,**f**).

## 5.4. Erosion Resistance

Limited investigations have been published on the influence of growth defects on material removal (erosion) when hard particles impact the hard coating surface [82,83]. It was found that the material removal occurred by repeated ductile indentation and cutting of the surface by impacting particles. Such solid particle erosion wear is the characteristic degradation of components in aircraft engines (operating in harsh environments), wind turbines, and power generation gas turbines. In order to enhance the higher reliability and longer lifetime of such components, many different hard protective coatings have been developed. The erosion resistance of monolithic coatings can be enhanced by using materials with both high hardness to inhibit crack initiation and high toughness to prevent crack growth. In contrast to monolithic hard coatings, various nanostructure coatings (e.g., TiN/TiAlN) are more appropriate for erosion wear protection because of their very high hardness and ability to inhibit crack propagation. Growth defects and other irregularities (scratches, pits) in the hard coating facilitate the crack initiation, which leads to premature breakup of the coating.

Wang et al. [82] performed a low-angle slurry erosive test of CrN/NbN superlattice coatings and observed the selective wear at defects. Similar wear mechanisms, like in the case of erosion with hard particles ejected from the nozzle at high pressure of air or liquid, were observed for components exposed to cavitation in different liquid media [84]. Cavitation implies the build-up and subsequent implosion of bubbles. Cavitation bubbles collapse violently either in the form of micro-jets or shock waves of high velocities, pressures, and temperatures. Due to cyclic impact of imploding cavitation bubbles on the solid surface, cavitation damage appears. Surface topography significantly influences the cavitation behavior because the implosion of bubbles is promoted especially at surface irregularities. During the cavitation erosion test, some nodular defects or droplets are removed from the surface

of the coating, leaving cavities behind. Such cavities can act as crack initiation sites. Azar et al. [83] studied the cavitation erosion of TiN coating, produced by arc-PVD and found that droplet-related defects have an important influence on the cavitation erosion resistance of the coating. During the cavitation erosion test, deep cavities were formed by the detachment of conical droplets. Such localized coating damages depend on the shape, position, and depth of the droplets in the coating.

### 5.5. Corrosion Resistance

Not only the tribological properties, but also the corrosion properties are affected by growth defects. It is well known that any kind of porosity causes a pitting corrosion. Porosity can be either a macroporosity or a microporosity. The former arises from large growth defects such as detached droplets, nodular and flake defects. The latter is determined by the growth morphology itself (e.g., open columnar structures arising from insufficient adatom mobility).

Small microstructural defects (e.g., pinholes, pores, and cracks) formed during or after deposition of PVD hard coatings act as channels for the corrosion of the substrate [30–34,85–87]. Therefore, such coating imperfections limit its protective nature in corrosive media. When the coated substrate is immersed in a corrosive medium, the electrolyte penetrates to the substrate (driven by capillary forces) through any pinholes extending down to the substrate. This leads to the formation of local galvanic corrosion between the substrate (acting as the anode) and the coating (acting as the cathode). Thus, anodic dissolution of the exposed substrate area occurs. The corrosion attack is more intensive in the case of a less noble substrate. Transition metal nitride coatings, which are the common choice for tool wear protection, are chemically more electronegative than steel. The corrosion medium in the pores is fast due to the large ratio of the cathode to anode areas. Pits form and extend radially from the pores, resulting in the cracking and removal of the upper coating by flaking. SEM and FIB images (Figure 29) show two examples of typical pitting corrosion for TiAlN hard coating deposited on a D2 tool steel substrate. A corrosion test was performed in a chlorine solution using electrochemical impedance spectroscopy. Both samples were exposed to corrosion medium (0.5 M NaCl; pH = 3.8) for 96 h.

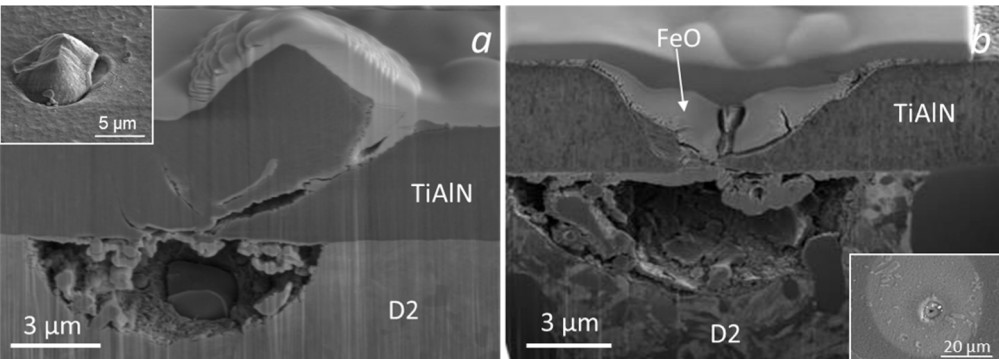

**Figure 29.** SEM top view (inset) and FIB images of a nodular defect (**a**) and a pinhole (**b**) in the sputter-deposited TiAlN hard coating exposed to corrosion medium (0.5 M NaCl; pH = 3.8, 96 h). An intensive pitting corrosion occurred at both sites.

Corrosion resistance of coated specimens can be improved if one could eliminate the growth defects in the coating. Although various techniques can be used to minimize the number of pinholes, they cannot be totally eliminated. Several approaches proposed to improve the corrosion resistance of PVD hard coatings were published in our previous paper [46]. One of the most promising ways is to combine the PVD coating and the thin atomic layer deposition (ALD) layer. ALD allows a deposition of conformal coating on non-even surfaces and coverage of particles, pinholes, and defects in the PVD coating.

### 5.6. Oxidation Resistance

High temperature oxidation resistance is one of the most important properties of PVD hard coatings for protection of cutting tools that are used for high speed, dry and hard machining. During such machining conditions the temperature at the cutting tool edge may reach more than 800 °C due to the high friction between the tool and the workpiece material. Thus, the protective coating suitable for advanced applications of cutting tools must endure extremely high thermo-mechanical loads and resist degradation in severe environments. Similar to corrosion, different coating imperfections have an important role during oxidation of PVD hard coatings. If the coated substrate is exposed to high-temperature oxidation, then the growth defects act as starting points for degradation and environmental attack (Figure 30). All the pores, voids, and gaps act as preferential diffusion paths for the oxygen transport to the inner coating/substrate interface and for metal ion transport from the substrate towards the surface.

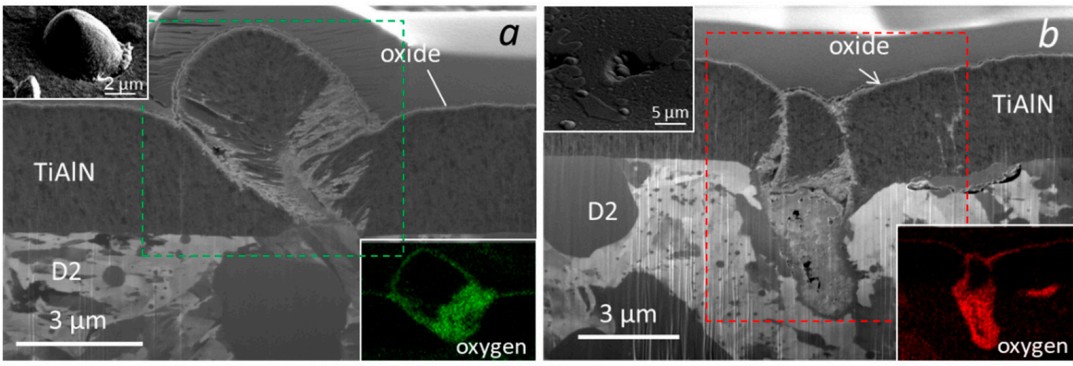

**Figure 30.** FIB and SEM top-view images of a nodular defect (**a**) and a pinhole (**b**) in the sputter-deposited TiAlN hard coating oxidized at 800 °C for 1 h. Oxygen elemental maps of area marked with the dashed frames are added in the inset.

There are only a few papers in the literature dealing with the role of growth defects during the oxidation process of hard coatings. The influence of defects on oxidation resistance of TiAlN is briefly mentioned in an article by McIntyre et al. [88]. The role of defects in the oxidation process is discussed in more detail by Lembe et al. [89]. They studied the influence of growth defects on the localized oxidation behavior of TiAlN/CrN superlattice coatings deposited by cathodic arc/unbalanced magnetron deposition on cemented carbide and HSS substrates. They found that oxidation behavior of the substrate material directly influences the generation of growth defects through craters caused by droplet formation. Some detached growth defects formed craters, through which oxidation products formed from the substrate material. Localized oxidation can also take place at the pores formed in the underdense region at the rim of the nodular defects. Two major kinds of oxides can emerge at the defects: oxides rich in Ti or oxides formed by substrate material. The formation of either one or the other probably depends on the depth of the defect and time of the heat treatment. Polcar and Cavalerio [90] investigated the thermal stability, oxidation resistance, and high temperature tribology of CrAlTiN coating deposited by cathodic arc evaporation on cemented carbide substrates. They found that the presence of surface defects caused oxidation of the cemented carbide substrate, which was particularly evident after the tribological tests at 700 and 800 °C. Hovsepian et al. [91] studied the influence of growth defects in deposited CrN/NbN nanostructured coatings on the high temperature corrosion resistance in a pure steam atmosphere. They found that the coating degradation mechanism when exposed to 650 °C in a pure steam environment is adverse diffusion of the substrate elements and oxygen through coating growth defects or cracks formed due to thermal expansion coefficient mismatch between the coating and the substrate. Fernades et al. [92] also demonstrated that the oxidation of TiSiVN coatings is controlled by the formation of a silicon oxide diffusion barrier which is affected by nodular defects in the as-deposited film. At nodular defects a complex oxide structure was

developed. Recently our research group published the results of our investigation on high temperature oxidation of nanolayered CrVN-based coatings [93]. We found that $V_2O_5$ patches started to appear around the nodular defects during high temperature oxidation.

Several approaches have been proposed to increase the oxidation resistance of PVD hard coatings. All approaches are based on how to prepare the coatings with a denser microstructure. There are various ways to achieve this. More dense coatings can be prepared, for example, by the HIPIMS deposition technique. On the other hand, the oxidation resistance of PVD hard coating can be improved by doping with different elements that cause the formation of a fine grain microstructure and segregation of the doping element at grain boundaries. More grain boundaries prolong the diffusion paths of oxygen and metal atoms. In addition, nanolayer coatings are more resistant to oxidation than single-layer ones. In nanolayer coatings many interlayer interfaces act as a diffusion barrier for inward diffusion of O and the outward diffusion of metal atoms.

*5.7. Gas Permeation*

Gas barrier coatings were first commercially applied on polymeric foil substrates for food packaging (since the early 1970s) [38,94]. Later their use was expanded to the pharmaceutical and beverage industry. An effective barrier layer can prevent losses (e.g., aroma) from the packaged product and prevent penetration into the package (oxygen, water vapor), both of which can affect product quality and expiration date. Aluminum metallized polymeric (mainly polyester) foil substrates are widely used for this purpose. In order to fulfill additional requirements, such as product visibility and microwaveability, transparent barrier coatings based on aluminum oxide or silicon oxide have been introduced.

Thermal and electron beam evaporation are the most frequently used deposition techniques to manufacture barrier films for food packaging. These evaporation techniques allow the preparation of films with a medium barrier performance, however, at a very high productivity. On the other side, reactive sputtering and plasma-enhanced chemical vapor deposition (PECVD) assure a significantly lower water vapor permeability, but at a lower productivity.

However, the permeability of coated foil substrates is not zero. The residual permeation and the effectiveness of PVD layers as gas diffusion barriers are attributed to the presence of microscopic defects in the coating. The gas transport through the barrier layer takes place mostly at pinholes. Therefore, the density and distribution of defects in the barrier film is a critical aspect for using barrier films.

Recently the application of thin film barrier coatings has expanded to flexible electronics (e.g., organic transistors, displays, thin film solar cells, organic light emitting diodes (OLEDs)) [95–98]. Flexible electronic devices are very sensitive to a reaction with water vapor and oxygen, and therefore require an encapsulation to protect against degradation. In order to reduce the rate of permeation of gases and vapors through polymer substrates, several barrier options have been proposed and utilized. One has to be aware that not only the barrier layers need to have a low defect density, but also the substrate particles and defects need to be covered and planarized.

Commonly two approaches fulfill the specification of water vapor and oxygen permeation. One approach is the optimization of single layers by using the atomic layer deposition technique which offers the deposition of perfect, high density, uniform, and conformal barrier layers on non-even surfaces and coverage of particles and defects on the substrate surface. The other approach is the deposition of multilayer stacks [99]. In a multilayer stack two or more inorganic barrier layers (e.g., $SiO_2$, $Si_3N_4$, $Al_2O_3$) are combined with a polymer interlayer. The barrier layers provide a low water and oxygen permeation, while the interlayer planarizes and decouples pinhole defects as well as increases the diffusion path length of the permeating gas through the barrier layer. Magnetron processes are exclusively used to deposit the complete multilayer stack.

### 5.8. Wettability of Surfaces

Wettability of a liquid on a coating surface depends on the surface chemistry, surface topography (asperities, nodular defects) as well as the liquid used. Applications where solid surface wettability plays a crucial role include contact lenses, body implants, biofilm growth, super-hydrophilic surfaces, self-cleaning, and nonstick surfaces. Wettability can be estimated by measuring the contact angle of the substrate with a given liquid. The balance at the three-phase contact of solid, liquid, and vapor is given by the well-known Young equation. The Young equation assumes that the surface is chemically homogeneous and topographically smooth. However, this is not true in real surfaces like PVD coatings. Instead of one equilibrium contact angle, a range of contact angles exist, because the actual contact angle is the angle between the tangent to the liquid–vapor interface and the actual local solid surface. Therefore, in principle, the coating roughness enhances the wettability.

We investigated how the nodular defects present in the coated steel substrates influence the wettability of liquids. The static contact angle was measured by observation of a drop of a test liquid (deionized water, diiodomethane) on coated samples. It is valid that the lower the contact angle, the greater the tendency for the liquid to wet the solid surface energy. We measured the contact angles of the as-deposited nl-TiAlN/TiN hard coating and after gently polishing with diamond paste. How this changed the topography of the surface is shown the 3D-profilometer images in Figure 31. While the highest peaks (nodular defects) decreased, the lowest ones completely disappeared. The wettability measurements showed that the contact angles decreased significantly (for about 20%) if deionized water and $CH_2I_2$ (diiodomethane) were used, while the corresponding surface free energy increased more than 10%.

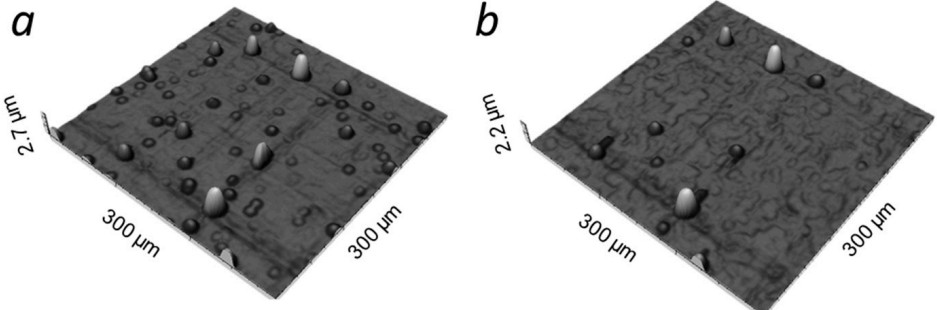

**Figure 31.** Three-dimensional (3D)-profilometer image of the same coating surface area before (**a**) and after polishing (**b**) with diamond paste. The sharp peaks are the nodular defects (pay attention to the strong exaggeration in z-scale).

## 6. Summary

We identified the origins of the growth defects found in PVD coatings prepared by three different deposition techniques. All topographic irregularities and foreign particles generated during different steps of substrate pretreatment and during the coating deposition process were systematically followed because they are seeds for the formation of growth defects. We need to be aware that contamination of substrates with foreign particles strongly depends on the quality of their wet cleaning procedure. Therefore, proper wet cleaning procedures, as well as an appropriate transport and batching, must be used in order to ensure a substrate surface that is as clean as possible.

Due to the inhomogeneity of the substrate material, certain topographic changes occur at sites of various inclusions already during polishing, as a result of the difference in their hardness in comparison with the matrix. A similar effect arises during the ion etching because the sputtering rates of various phases are also different. Whether a hole or protrusion will appear at the site of the inclusion depends on the polishing (removal) rate and etching rate. Special attention was given to the various ion etching

techniques used in industrial hard coating deposition systems (DC etching, MF etching, booster etching). We were particularly interested in how they affect the substrate topography.

It was found that a large part of the seed particles responsible for growth defect formation are generated during the coating process itself. The origin of such seed particles which can take many forms are described in more detail. The emphasis is on the seed particles that are characteristic for three different evaporation techniques used in our lab: thermionic arc evaporation, magnetron sputtering, and cathodic arc evaporation. In the PVD coatings prepared by the cathodic arc deposition process, the growth defects originate mostly from droplets formed during the coating deposition. Droplet-like defects are significantly smaller than others, but their density is two orders of magnitude higher. Therefore, the roughness of such coatings is relatively high.

The primary origin of growth defects in magnetron sputtering is the seed particles formed on the target surface outside of the racetrack due to the re-deposition and poisoning effect. At the early stage of deposition all these particles are charged and therefore accelerated away from the target due to the repulsion effect. Most of them reach the substrate surface and they become the seeds for growth of nodular defects in the coating.

The lowest concentration of growth defects was found in the coatings prepared by the thermionic arc evaporation. In contrast to both deposition techniques mentioned above, in this case the evaporation source is not the origin of larger quantities of seed particles.

A common form of growth defects for all three types of coatings is also flake defects. They originate from flakes, which are formed from a thick deposit on the shields and other components of the vacuum system that break off due to the high compressive stresses. Flaking of the coating is also triggered by arcing on the substrate turntable during ion etching and deposition process.

An important part of nodular defects is formed from wear particles that are generated by all the moving parts inside the vacuum chamber. Therefore, less moving parts in the fixture systems mean less defects.

The consequences of growth defect formation on functional properties of thin film are discussed in more detail. It is explained how growth defects affect the quality of optical coatings and thin layers for the production of semiconductor devices together with their role in tribological contacts as well as in corrosion and oxidation processes. The effect on the permeation and wettability of the coatings is also briefly described. The findings of other researchers are supported with the relevant results obtained in our lab.

**Author Contributions:** P.P., conceptualization, methodology manuscript writing; A.D., FIB analysis of growth defects and tribological tests; P.G., SEM and FIB analysis of non-metallic inclusions; M.Č., manuscript review and project administration; M.P., critical review and participation in the writing of the first two chapters. All authors have read and agreed to the published version of the manuscript.

**Funding:** This work was supported by the Slovenian Research Agency (program P2-0082, projects L2-4173, L2-4239, L2-4249, L2-5470). We also acknowledge funding from the European Regional Development Funds (CENN Nanocenter, OP13.1.1.2.02.006) and the European Union Seventh Framework Programme under Grant Agreement 312483—ESTEEM2 (Integrated Infrastructure Initiative—I3).

**Acknowledgments:** The authors would also like to thank Tonica Bončina and Gregor Kapun for SEM and EDS analyses and Jožko Fišer for performing some laboratory tests.

**Conflicts of Interest:** The authors declare no conflicts of interest. The funders had no role in the design of the study; in the collection, analyses, or interpretation of data; in the writing of the manuscript, or in the decision to publish the results.

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
