# Peer review of "Review of Growth Defects in Thin Films Prepared by PVD Techniques"

_coatings, doi:10.3390/coatings10050447_

Round 1

Reviewer 1 Report

This paper well summarized the research related to the growth defects in PVD coatings in decades. The types of growth defects, the evolution process of growth defects in different PVD techniques were introduced. The effects of growth defects on the quality of optical properties, property of semiconductor devices, friction and wear, corrosion, and oxidation resistance etc. were summarized . It gives a full image of the researches on the growth defects in PVD coatings and also very detailed information in each siginificant aspect related to growth defects. This work is very useful for these researchers who are working in the field. Please very carefully to check the wrrting format of this manuscript. At least following parts have formating or edition problem: Figure 2, line 323-325,341-344,378-379,386-390,616-618,697,789,979.

Reviewer 2 Report

The paper is devoted to surface defects formed in PVD coatings.

Mechanisms responsible for the defect formation, like surface pits and protrusions, and particles of different origins are reviewed in detail, with most examples given from PVD coating of steel alloys.

All main steps of the substrate processing - from surface grinding and wet cleaning to the PVD process itself are analyzed in terms of possible defect formation.

Many practical recommendations, recipes and "good practices" for avoiding defect formation are given, making this paper useful also for the introduction to PVD coating practice.

Influence of the defects on coating properties like corrosion and oxidation strength, wear resistance, gas permeability is also briefly reviewed.

The paper should be interesting for all researchers whose activities are related to coating, from tool steel coating to semiconductor metal and dielectric layer deposition.

The paper can be published in the present form.

A minor technical correction:

Link to reference [4] is missing in the 112 after "electrodeposition".

Reviewer 3 Report

In the manuscript, authors gave a nice overview of growth defect in thin films prepared by three different PVD techniques (electron beam evaporation, magnetron sputtering and cathodic arc evaporation). They discuss detects formation during three preparation steps (mechanical pretreatment of the substrate, substrate ion etching and deposition process). They were mainly focused on the deposition of various type of (TiAlN/CrN) nanolayers onto three different stainless steel substrate.

The manuscript is well structured. In the Introduction chapter authors gave a nice historical overview of growth defects in thin-film materials investigation. Then in the second and third chapter follows an overview of surface regularities from substrate pretreatment and growth defects formed during deposition. The fourth chapter is dedicated to the origin of seed particles. In the final chapter, they have discussed the influence of growth defects on functional properties of thin films.

Content and obtained results are very interesting and expressed in proper English language. Especially interesting and valuable is a set of SEM images of samples surface morphology and FIB/SEM cross-section images that support the classification of growth defects.

But the text is not properly formatted and it contains many typing errors (few of them are listed below). The text should be carefully checked and corrected.

Most of the presented content is very nicely supported by references, but there are several subsections where statements are not supported by references. For example, several statements about substrate mechanical pretreatment (row 67-74), following by ion etching and PVD growth defects. Please add relevant references for that part and also for the rest of the manuscript.

In the manuscript are presented SEM images of several different groups of samples prepared by different techniques. It is not clear whether images are taken from authors other publication. Also, more details about analysed samples are missing (preparation, etc.). At least reference to other publication should be given. It’s confusing that somewhere are analysed and discussed TiAlN and somewhere TiN samples (for example Fig. 9).

Other comments and suggestions that can improve the quality of the manuscript:

  • Row 27, Page not presented “Napaka. Zaznamek ni definiran.”
  • Row 212: space missing after Figure 1
  • Fig 1: It's not clear how defects are detected and recognized by EDX from only five very large areas. This should be explained in more details.
  • Row 215-217 “three different types of substrates (D2, powder metallurgical (PM) ASP30 and stainless steel 316L). ”This statement is not in agreement with results presented at Fig 1 and in Table 1 where are presented results for only two type of samples.
  • Table 1: Please correct rounding/formating of calculated values to two or one significant digits. 
  • Figure 2: AFM images scale labels are not readable (quality of figures should be improved). SiO2 label “2” should b in subscript. What are white trapezes in fig a and b?
  • Fig 2 two times added to the text. 
  • Row 268: “nanoilayer” typing error?
  • Row 288-293 References missing 
  • Row 297-299 Reference for cleaning procedure missing
  • Row 312-314 Reference missing for ion etching influence on adhesion and of the coating.
  • Row 318 Reference missing for statement:”Substrate surfaces which are exposed to an ion etching erode and change topography. In general, surface topography depends on the time of etching, density of the plasma, energy of ions and type of ions. Different morphological features like cones, pits, hillocks, pyramids are formed and their formation is closely related to the initial surface irregularities, impurities and variations in the sputtering yield as a function of the angle of the ion beam incidence to the surface.”
  • Row 322-326 text formating should be corrected.
  • Row 379: Reference to Figure should be corrected “Figure6Figure1”
  • Row 386 Fig. 2 again?
  • Figure 7b Not clear caption
  • Figure 8: resolution should be improved. Scale labels not readable. I’m suggesting to do line profile in each AFM image for comparison (difference in height in z direction). Units of z scale missing
  • Figure 9c: What is H11?
  • Figure 10: What is HM?
  • Figure 11: Missing scalebar label (length).
